# Dynamic wind farm flow control using free-vortex wake models

**Maarten J. van den Broek**[1], **Marcus Becker**[1], **Benjamin Sanderse**[2], **and Jan-Willem van Wingerden**[1]

[1]Delft Centre for Systems and Control, TU Delft, Mekelweg 2, 2628CD Delft, NL
[2]Scientific Computing, CWI, P.O. Box 94079, 1090GB Amsterdam, NL

**Correspondence:** Maarten J. van den Broek (m.j.vandenbroek@tudelft.nl)

**Abstract.** A novel dynamic economic model-predictive control strategy is presented that improves wind farm power production and reduces the additional demands of wake steering on yaw actuation when compared to an industry state-of-the-art reference controller. The novel controller takes a distributed approach to yaw control optimisation using a free-vortex wake model. An actuator-disc representation of the wind turbine is employed and adapted to the wind-farm scale by modelling secondary effects of wake steering and connecting individual turbines through a directed graph network. The economic model-predictive control problem is solved on a receding horizon using gradient-based optimisation, demonstrating sufficient performance for realising real-time control. The novel controller is tested in a large-eddy simulation environment and compared against a state-of-the-art look-up table approach based on steady-state model optimisation and an extension with wind direction preview. Under realistic variations in wind direction and wind speed, the preview-enabled look-up table controller yielded the largest gains in power production. The novel controller based on the free-vortex wake produced smaller gains in these conditions, while yielding more power under large changes in wind direction. Additionally, the novel controller demonstrated potential for a substantial reduction in yaw actuator usage.

## 1 Introduction

Wind farm flow control aims to improve wind turbine performance by reducing aerodynamic wake interaction between turbines which are often placed in large, densely spaced wind farms to effectively make use of limited available space (van Wingerden et al., 2020). Strategies such as wake redirection through yaw misalignment and dynamic induction control with blade pitch variations have been shown to achieve improvements in power production and reductions in fatigue loading (Meyers et al., 2022).

Wake redirection makes use of intentional yaw misalignment to steer wakes away from downstream turbines. When effectively applied, a small power loss is incurred on the upstream wind turbine which results in a larger power gain on the downstream turbine. This has been demonstrated in wind tunnel experiments (Campagnolo et al., 2016; Bastankhah and Porté-Agel, 2019; Campagnolo et al., 2020) and several field studies (Howland et al., 2019, 2022; Fleming et al., 2020, 2021; Doekemeijer et al., 2021; Simley et al., 2021a).

The control strategies to apply wake steering in wind farms may be roughly divided into model-based and model-free approaches. The latter attempts to synthesise control signals directly from measurements of the wind farm. In wind tunnel experiments, a closed-loop, model-free yaw controller (Campagnolo et al., 2016) and extremum-seeking control (Kumar et al., 2023) have been demonstrated to produce power gains from wake steering under steady flow conditions. Extremum-seeking control has also been demonstrated in large-eddy simulation (LES) (Ciri et al., 2017). These data-driven methods have not been tested under realistic time-varying wind direction variations. To improve interpretability of these methods, Sengers et al. (2022) introduces a purely data-driven wake model with physically explainable parameters. However, it still requires wake measurements which are not generally available in the field.

Recent work on wake steering uses a model-based approach that embeds prior knowledge and allows better generalisation to different operating conditions. The steady-state models in the FLORIS toolbox (NREL, 2022), such as the cumulative curl (Martínez-Tossas et al., 2019) and Gauss-

curl hybrid (King et al., 2021) models, provide an approximation for the time-averaged velocity profiles in the wake. These models allow efficient optimisation of steady-state optimal yaw angles for wake steering to generate look-up tables (LUT) with yaw offsets for varying wind directions. These LUT approaches have been used, for example, for yaw control under steady conditions in LES (Gebraad et al., 2016), in a wind tunnel setting with simulated wind direction changes (Campagnolo et al., 2020), and in a closed-loop control framework with model adaptation under time-varying inflow in LES (Doekemeijer et al., 2020). Howland et al. (2022) most recently demonstrates the effective use of a tuned steady-state model for wake steering in a field experiment.

However, the validity of steady-state models may be limited under realistic, time-varying inflow conditions. The inclusion of wake dynamics is essential for active power control in wind farms (Shapiro et al., 2018) and the dynamics of realistic wind direction variations need to be accounted for in control optimisation (Kanev, 2020). For that purpose, some studies have adapted the steady-state engineering wake models to include dynamics (Lejeune et al., 2022; Becker et al., 2022b; Branlard et al., 2023) or investigated wind direction preview to account for the dynamics of wake propagation (Simley et al., 2021b; Sengers et al., 2023). On the other hand, a physics-based approach may naturally include the dynamics of wake propagation. The use of LES for control optimisation showed promising results (Munters and Meyers, 2018) and recent work has approached real-time control by coarsening mesh resolution and adjusting control parameters (Janssens and Meyers, 2023). An approximation of wind farm flow using two-dimensional computational fluid dynamics (Boersma et al., 2018; van den Broek and van Wingerden, 2020) has been attempted and proven useful for induction control (van Wingerden et al., 2017; Vali et al., 2019), but inherently lacks the wake dynamics required to capture the wake deflection under yaw misalignment (van den Broek et al., 2022b).

A dynamic, control-oriented free-vortex wake model (FVW) of the wind turbine wake was developed for gradient-based control optimisation and shown to capture sufficient wake flow dynamics to model wake deflection for control (van den Broek et al., 2022a). The economic model-predictive control implementation yielded promising results for wake steering under time-varying inflow conditions. The model formulation based on Lagrangian particles allows greater flexibility compared to mesh-based flow calculations (van den Broek et al., 2023b). Additionally, the model has been validated for power predictions for wind turbines operating under yaw misalignment (van den Broek et al., 2023a). Despite its flexibility, the optimisation with the FVW is currently limited to single wakes by the stability of the free-vortex methods and the exponential increase in computational complexity with larger numbers of vortex elements.

To extend economic model-predictive control with the FVW to larger wind farms, this paper develops a distributed approach to control optimisation for wake steering under time-varying inflow conditions. The performance of the novel control strategy will then be evaluated in LES against the greedy control baseline, and, more importantly, a reference controller based on the industry state-of-the-art use of a LUT with steady-state optimised yaw offsets as well as an extension with wind direction preview. In addition to synthetic wind signals, a set of measured wind direction and wind speed variations will be used to evaluate performance in a simulated section of the Hollandse Kust Noord (HKN) wind farm.

The contribution of this paper is twofold: (i) development of a distributed approach to dynamic economic model-predictive control for wake steering with a free-vortex wake model, (ii) validation of the control strategy under realistic, turbulent inflow conditions with wind direction and wind speed variation.

The remainder of this paper is structured as follows. Section 2 introduces the FVW model for the wind turbine wake and the coupling to facilitate farm-scale optimisation. The model-predictive control strategy is developed in Sect. 3. The reference controllers and simulation test cases for validation are defined in Sect. 4. The results are then discussed in Sect. 5 and, finally, the conclusions are shown in Sect. 6.

## 2   Model Development

The core of the novel dynamic model-predictive control strategy is the FVW model, briefly described in Sect. 2.1. In order to implement this model in a farm-scale controller, Sect. 2.2 presents a strategy for incorporating secondary steering effects when a turbine operates in the wake of a yaw-misaligned turbine. Section 2.3 then illustrates the strategy for connecting wind turbines into wind farms by constructing a directed graph connecting upstream and downstream neighbouring turbines.

### 2.1   Wake model for control optimisation

The wake model used for yaw control optimisation is an actuator-disc representation of the wind turbine modelled with the free-vortex wake as developed in (van den Broek et al., 2022a) and validated for power predictions for wake steering control with yaw misalignment (van den Broek et al., 2023a), which yielded the current model parameters listed in Table 1. The model, illustrated in Fig. 1, assumes a uniformly loaded actuator disc that sheds vorticity from its edge. These rings of vorticity are discretised in straight-line vortex filaments and advected downstream as Lagrangian particles, forming a skeletal representation of the wind turbine wake.

A non-linear state-space system is defined for the model dynamics which updates the model state vector $q_k \in \mathbb{R}^{n_s}$,

**Table 1.** Numerical parameters for the FVW actuator-disc model.

| | | |
|---|---|---|
| time step | $\Delta t \cdot u_\infty / D$ | 0.3 |
| number of rings | $n_\mathrm{r}$ | 40 |
| elements per ring | $n_\mathrm{e}$ | 16 |
| initial core size | $gma/D$ | 0.16 |
| turbulent growth | $\delta$ | 100 |
| yaw exponent - thrust | $\beta_\mathrm{t}$ | 1 |
| yaw exponent - power | $\beta_\mathrm{p}$ | 3 |

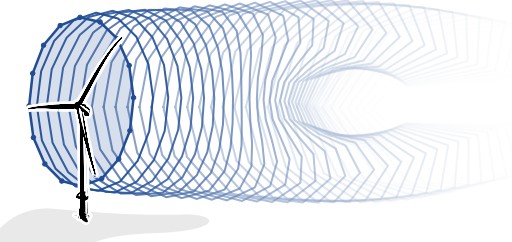

**Figure 1.** Free-vortex wake model of the wind turbine wake. Rings of vorticity discretised in straight-line vortex filaments are shed from the wind turbine rotor modelled as an actuator disc forming a skeletal representation of the wake. The wake develops the characteristic curled shape for turbine operation under yaw misalignment.

with the number of states $n_\mathrm{s}$, at discrete time-step $k$ as

$$\boldsymbol{q}_{k+1} = f(\boldsymbol{q}_k, \psi_k, a_k, \boldsymbol{u}_\infty), \tag{1}$$

where the state vector contains the start and end points, vorticity, and core size for all vortex filaments. The turbine yaw heading $\psi_k$ and the induction factor $a_k$ are control inputs and $\boldsymbol{u}_\infty$ is the free-stream velocity. The yaw misalignment angle $\gamma = \theta - \psi$ is the difference between turbine heading $\psi$ and wind direction $\theta$.

At fixed time intervals $\Delta t$, a vortex ring discretised in $n_\mathrm{e}$ vortex filaments is generated at the edge of the rotor. At the same time, a vortex ring at the end of the wake is removed to maintain a finite wake with $n_\mathrm{r}$ rings. The vorticity $\Gamma$ generated along the edge of an actuator disc is directly related to the pressure differential generated by the disc (van Kuik, 2018),

$$\Gamma = \Delta t \frac{\partial \Gamma}{\partial t} = \Delta t \frac{1}{\rho} \frac{T}{A_\mathrm{r}}, \tag{2}$$

where $\rho$ is the air density, $A_\mathrm{r}$ is the area swept by the rotor, and $T$ is the thrust force. The vortex filaments are convected over time with a rate $\dot{\boldsymbol{x}} \in \mathbb{R}^3$

$$\dot{\boldsymbol{x}} = \boldsymbol{u}_\mathrm{ind}(\boldsymbol{x}) + \boldsymbol{u}_\infty(\boldsymbol{x}), \tag{3}$$

which is the combination of the free-stream velocity $\boldsymbol{u}_\infty \in \mathbb{R}^3$ and the total velocity induced by all filaments $\boldsymbol{u}_\mathrm{ind} \in \mathbb{R}^3$ at the vortex position $\boldsymbol{x} \in \mathbb{R}^3$.

The induced velocity of an individual vortex filament $\boldsymbol{u}_\mathrm{i} \in \mathbb{R}^3$ at a point $\boldsymbol{x}_0 \in \mathbb{R}^3$ according to the Biot-Savart law (Katz and Plotkin, 2001; Leishman, 2000),

$$\boldsymbol{u}_\mathrm{i}(\boldsymbol{x}_0) = \frac{\Gamma}{4\pi} \frac{\boldsymbol{r}_1 \times \boldsymbol{r}_2}{||\boldsymbol{r}_1 \times \boldsymbol{r}_2||^2} \boldsymbol{r}_0 \cdot \left( \frac{\boldsymbol{r}_1}{||\boldsymbol{r}_1||} - \frac{\boldsymbol{r}_2}{||\boldsymbol{r}_2||} \right), \tag{4}$$

where the relative positions $\boldsymbol{r} \in \mathbb{R}^3$ for a vortex filament starting at $\boldsymbol{x}_1 \in \mathbb{R}^3$ and ending at $\boldsymbol{x}_2 \in \mathbb{R}^3$, with vortex strength $\Gamma$, are defined as

$$\boldsymbol{r}_0 = \boldsymbol{x}_2 - \boldsymbol{x}_1, \tag{5}$$
$$\boldsymbol{r}_1 = \boldsymbol{x}_1 - \boldsymbol{x}_0, \tag{6}$$
$$\boldsymbol{r}_2 = \boldsymbol{x}_2 - \boldsymbol{x}_0. \tag{7}$$

A Gaussian core with core size $gma$ is included to regularise singular behaviour of the induced velocity close to the vortex filament,

$$\boldsymbol{u}_{\mathrm{i},gma}(\boldsymbol{x}_0) = \boldsymbol{u}_\mathrm{i}(\boldsymbol{x}_0) \left( 1 - \exp \left( -\frac{||\boldsymbol{r}_1 \times \boldsymbol{r}_2||^2}{gma^2 ||\boldsymbol{r}_0||^2} \right) \right). \tag{8}$$

The effects of turbulent and viscous diffusion are approximated using the growth of the vortex core

$$gma_{k+1} = \sqrt{4\alpha\delta\nu\Delta t + gma_k^2}, \tag{9}$$

which is Squire's modification of the diffusive growth of the Lamb-Oseen vortex core (Squire, 1965), with the discrete time step $k$, the constant $\alpha = 1.25643$, effective turbulent viscosity coefficient $\delta$ to tune core growth, kinematic viscosity $\nu = 1.5e - 5 m^2.s^{-1}$, and time step $\Delta t$.

The thrust $T$ and aerodynamic power $p_\mathrm{a}$ for the rotor model are calculated as

$$T = c_\mathrm{t} \cdot \frac{1}{2}\rho A_\mathrm{r} u_\infty^2 \cos^{\beta_\mathrm{t}}(\gamma), \tag{10}$$

$$p_\mathrm{a} = c_\mathrm{p} \cdot \frac{1}{2}\rho A_\mathrm{r} u_\infty^3 \cos^{\beta_\mathrm{p}}(\gamma), \tag{11}$$

where $c_\mathrm{t}$ and $c_\mathrm{p}$ are, respectively, the thrust and power coefficient and $u_\infty$ is the magnitude of the free-stream inflow velocity. For performance evaluation in terms of available power for downstream turbines, the free-stream velocity $u_\infty$ in (11) is replaced by the rotor-averaged velocity $u_\mathrm{r}$ at the position of the downstream rotor, which includes the velocity deficit from the aerodynamic wake. This rotor-averaged velocity is calculated as

$$u_\mathrm{r} = \left|\left| \frac{1}{n_\mathrm{u}} \sum_{i=1}^{n_\mathrm{u}} \boldsymbol{u}_\infty(\boldsymbol{p}_i) + \boldsymbol{u}_\mathrm{ind}(\boldsymbol{p}_i) \right|\right|_2, \tag{12}$$

where $n_\mathrm{u}$ sampling points $\boldsymbol{p}_i \in \mathbb{R}^3$ are evenly distributed over the rotor area.

The yaw dependence of the coefficients can be tuned with the cosine exponents $\beta_\mathrm{t}$ and $\beta_\mathrm{p}$ for thrust and power, respectively, such as seen in Hulsman et al. (2022). The current values for these exponents are based on work by van den Broek

et al. (2023a), although they may differ in reality (Howland et al., 2020; Li and Yang, 2021) and, thus, may require tuning for different turbine types or atmospheric conditions. Additionally, a dependence on thrust force (Heck et al., 2023) or on wind field heterogeneity (Liew et al., 2020) is not included in the current work.

The induction factor is used to calculate the thrust coefficient and power coefficient for the model as

$$c_{\mathrm{t}}(a) = \begin{cases} 4a(1-a) & \text{if } a \le a_{\mathrm{t}}, \\ c_{\mathrm{t1}} - 4(\sqrt{c_{\mathrm{t1}}}-1)(1-a) & \text{if } a > a_{\mathrm{t}}, \end{cases} \quad (13)$$

$$c_{\mathrm{p}}(a) = 4a(1-a)^2, \quad (14)$$

with parameter $c_{\mathrm{t1}} = 2.3$ and the induction at the transition point $a_{\mathrm{t}} = 1 - \frac{1}{2}\sqrt{c_{\mathrm{t1}}}$ (Burton et al., 2001). In the current study, the induction factor is fixed to the optimum value known from momentum theory, $a = 0.33$, however it may also be used as a degree of freedom for induction control or to adapt the model to above-rated operating conditions.

## 2.2    Modelling secondary steering

One important effect that is not immediately accounted for in the FVW is the cumulative effect of wake steering. Wind turbines in the wake of a yaw-misaligned turbine need to yaw less to achieve the same wake deflection as an isolated turbine, as shown in simulation (Fleming et al., 2018) and wind tunnel experiments (Bastankhah and Porté-Agel, 2019). This cumulative effect of wake deflection is attributed to crossflow on the waked rotor and the trailing vortices from the yaw-misaligned turbine. The secondary steering effects have been accounted for in a control-oriented model in FLORIS by calculation of an effective yaw angle (King et al., 2021).

A simulation study is used to develop a method for incorporating these secondary steering effects in the current wake model. The study is performed with LES using settings as described in Sect. 4.6. The turbulent inflow has an average speed of $9\,\mathrm{m.s}^{-1}$. The effects of yaw misalignment are measured for 1, 2, 3, and 5 turbines with a $5\,D$ inter-turbine spacing, where $D$ is the rotor diameter. The layout is aligned with the wind direction. For three, and fewer, turbines, the domain size is $4km \times 2km \times 1km$. The five-turbine test is performed on a $6km \times 2km \times 1km$ domain. Cross-stream flow slices are recorded at $1\,D$ intervals downstream from the first turbine. The wake deflection, illustrated in Fig. 2, is calculated based on the average flow over the final $1500\,\mathrm{s}$ of the $2000\,\mathrm{s}$ simulations.

Based on these simulation results, we present a method for calculation of an induced yaw angle which is used to propagate the effects of secondary steering to downstream turbines with minimal additional complexity. It differs from the solution proposed by King et al. (2021) because the induced yaw effects are calculated directly from the sampled velocity.

For downstream neighbours, the velocity is sampled over a rotor-disc area. The effective flow direction $\theta_{\mathrm{eff}}$ is calculated

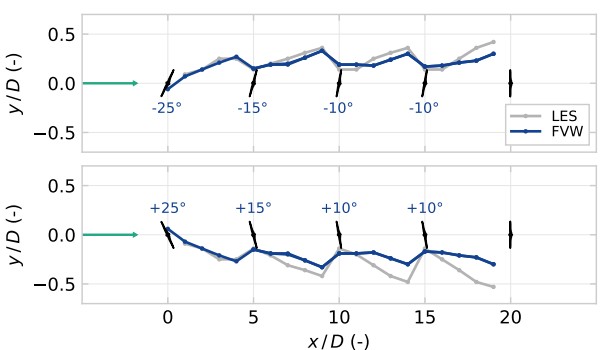

**Figure 2.** Deflection of the wake centre comparing the FVW modelling of induced yaw effects with time-averaged flow from LES. The individual FVW wakes have been combined for this comparison using a root-sum-square superposition of the wake deficit. The cumulative effect of wake steering is captured as a reduced yaw offset is required for similar levels of wake deflection when operating in the wake of yaw-misaligned turbines. The model is symmetric, whereas the LES data shows greater wake deflection from the second turbine onwards when implementing positive yaw misalignments.

from the velocity components in the horizontal plane. We take the root-mean-square of the wind direction $\boldsymbol{\theta}_{\mathrm{u}} \in \mathbb{R}^{n_{\mathrm{u}}}$ sampled over $n_{\mathrm{u}}$ points to get one effective flow direction,

$$\theta_{\mathrm{eff}} = \mathrm{RMS}(\boldsymbol{\theta}_{\mathrm{u}}). \quad (15)$$

The proposed induced yaw angle $\gamma_{\mathrm{ind}}$ is then the difference between the effective inflow and the nominal wind direction,

$$\gamma_{\mathrm{ind}} = \theta_{\mathrm{eff}} - \theta. \quad (16)$$

The optimised yaw offset $\gamma^\star$ is the result of the optimisation with the FVW model. The new induced yaw angle reduces this optimised yaw offset to yield the commanded yaw angle $\gamma_{\mathrm{ref}}$, which is sent to the wind turbine

$$\gamma_{\mathrm{ref}} = \gamma^\star - \gamma_{\mathrm{i}} \quad (17)$$

$$\text{with} \begin{cases} \gamma_{\mathrm{i}} = \max(\min(\gamma^\star, \gamma_{\mathrm{ind}}), 0) & \text{if } \gamma^\star > 0, \\ \gamma_{\mathrm{i}} = \min(\max(\gamma^\star, \gamma_{\mathrm{ind}}), 0) & \text{otherwise.} \end{cases}$$

The conditional application of the induced yaw ensures the yaw reference does not compensate for induced yaw to achieve zero offset.

Fig. 2 shows how this induced yaw angle contributes to approximating the secondary steering effects. The wake deflection is defined as the position where potential power from a virtual rotor placed in the stream would be minimal, as used in e.g. (Schottler et al., 2018; van den Broek et al., 2023a). The FVW results are based on individually simulated wakes which have been combined using root-sum-square superposition of the wake deficit. The induced yaw angles from the

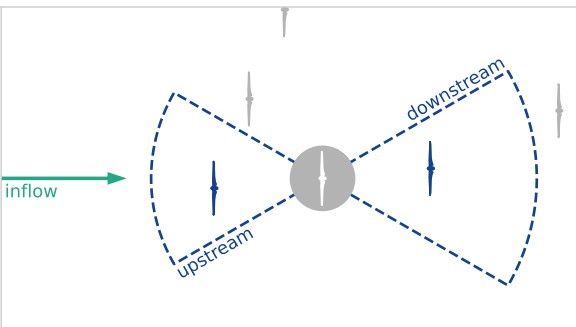

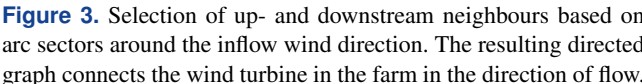

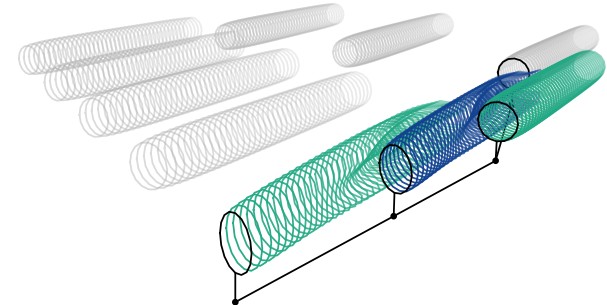

**Figure 3.** Selection of up- and downstream neighbours based on arc sectors around the inflow wind direction. The resulting directed graph connects the wind turbine in the farm in the direction of flow.

**Figure 4.** Representation of the wakes in a wind farm using a network of FVW models, with an indication of the graph connecting the wakes which have been individually simulated. The highlighted wakes show a wake and its immediate upstream and downstream neighbours. The upstream wake simulation provides induced yaw estimates for incorporating secondary steering in the control signal for the downstream turbine. The downstream neighbour is accounted for in the optimisation for wake steering to minimise negative effects from wake interaction.

first two upstream turbines for each turbine are added to the actual yaw misalignment with respect to the free-stream inflow. The downstream turbines operate at a smaller yaw offset magnitude, but achieve similar levels of wake redirection. This captures the secondary steering effects for implementation in the control optimisation strategy. Note that the induced yaw effects are not applied on turbines that are operating without yaw offset, as it would lead to unwanted offsets.

The downstream deflection from the second turbine onwards is captured better for negative yaw misalignments. Wake redirection with positive yaw offsets appears to lead to more deflection on downstream turbines in the LES simulations due to rotating flow in the wake and ground effects, but modelling this asymmetry is out of scope of this paper. In future work, an asymmetric thrust-yaw curve could be implemented or further refinements could be incorporated in a model adaptation stage in a closed-loop control implementation.

### 2.3 Directed graph network

The communication protocol between upstream and downstream neighbours is constructed based on a directed graph network, similar to, for example, the work by Starke et al. (2021). The structure of this network naturally changes with the wind direction as wakes propagate with the flow through the farm. The relevant neighbouring turbines are selected based on arc sectors around the wind turbine as illustrated in Fig. 3. The arc sectors are defined by a radius of influence and a spreading angle around the predicted inflow. Separate directed graphs are constructed for the upstream and downstream connection, although they may be symmetric.

The upstream graph is used for propagating the induced yaw effects to account for the secondary effects of wake steering. The downstream graph is used to determine which turbines are relevant in the optimisation for wake redirection control. Simulated wake length and prediction horizon are both important in determining suitable arc radius settings; downstream turbines, for example, should only be included in the optimisation problem if adequately covered by the simulated wake length and the prediction horizon. The spreading angle limits the connection to only those wakes that may actually interact through the streamwise wake propagation. It should be wide enough to cover the width of the wake and possible deflection due to yaw misalignment. An example network of FVW models is shown in Fig. 4 using a symmetric upstream and downstream graph, illustrating how the wake models are connected along the flow direction through the farm. controls.

## 3 Controller synthesis

In this section we develop an economic model-predictive wind farm controller around the network of FVW models. Section 3.1 describes a reduction of the dimensionality of the optimisation using a B-spline basis. The optimisation problem for the open-loop receding horizon control strategy is then defined in Sect. 3.2.

### 3.1 Basis functions for control signal

Previous work (van den Broek et al., 2022a, 2023b) uses a control signal that may be freely chosen at every simulation time step. However, the current implementation of the model uses forward-mode automatic differentiation for constructing the gradients for optimisation, as opposed to the manual derivation of the adjoint method developed by van den Broek et al. (2022a). The automatic differentiation framework yields additional flexibility in model development

and facilitates improvements in computational performance by minimising code complexity. Furthermore, it drastically reduces the memory requirements for gradient calculation compared to the manual adjoint derivation, which required storing all partial derivatives at every time step. As a trade-off, it comes with a computational cost that scales linearly with the number of control degrees of freedom. For that reason, the current work aims to limit the possible search space to improve optimisation performance.

The dimensionality of the problem is reduced by constructing the control signal using B-splines. For the optimisation, the control signal needs to be defined over a prediction horizon of $N_h$ steps from the current step $k = k_0$. The reference turbine yaw heading $\psi$ is calculated from a spline $s(k, \boldsymbol{c})$ defined on the range $k \in [k_0; k_0 + N_h]$ as

$$\psi_k = s(k, \boldsymbol{c}), \tag{18}$$

at time step $k$ with $n_b$ the number of B-spline basis functions with the corresponding coefficients $\boldsymbol{c} \in \mathbb{R}^{n_b}$. Fig. 5 illustrates the construction of a control signal from an example B-spline basis with $n_b = 7$ splines, starting at $k_0 = 0$ and with a prediction horizon $N_h = 80$ steps.

To further reduce the dimensionality, not all coefficients are left to be free variables in the optimisation problem. The first coefficient is chosen equal to the current yaw angle to ensure a continuous yaw signal, $c_1 = \psi_{k_0}$. The turnpike effect (Dorfman et al., 1958), also illustrated by van den Broek et al. (2022a), leads turbines to always return to greedy control towards the finite optimisation horizon. Therefore, in the example illustrated in Fig. 5, the final three coefficients, $c_5, c_6, c_7$, are chosen equal to the wind direction which leaves the remaining coefficients, $c_2, c_3, c_4$, free as the control parameters for the optimisation problem.

The smoothness of the B-spline basis improved the behaviour of the gradient for optimisation with the FVW in trial optimisations. The basis functions average out noisy contributions to the gradient and smoothen the optimisation landscape. This allows the optimisation problem in the current work to be defined with a lower input regularisation cost while still yielding smooth control signals. The dimensionality reduction from the use of basis functions does limit some of the flexibility in the control solutions that can be found compared to fully free optimisation.

## 3.2 Distributed optimisation

In order to scale the model-based control approach with the FVW to the wind-farm scale, a distributed approach is implemented as illustrated in Fig. 6 and described in Alg. 1. In this approach, each individual turbine has its own wake model. The optimisations for all turbines are then performed in parallel, where each of the turbines attempts to optimise its control signal considering wake effects on its immediate downstream neighbours given an expected inflow over the prediction horizon that is kept fixed during the iterations of the non-

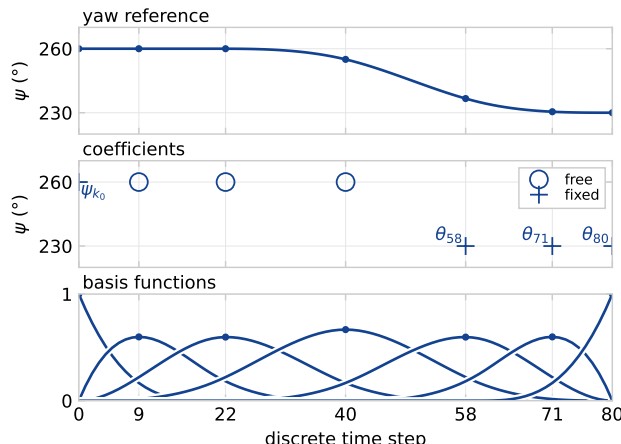

**Figure 5.** B-spline basis with $n_b = 7$ splines for constructing a yaw control signal over the prediction horizon. The first coefficient is fixed to the preceding yaw reference for continuity, $c_1 = \psi_{k_0}$. The final three coefficients, $c_5, c_6, c_7$, are set equal to the wind direction $\theta$ at the associated time steps because the optimisation returns to greedy control towards the finite horizon. The remaining three coefficients, $c_2, c_3, c_4$, are free in the optimisation.

linear solver. This is an economic model-predictive control problem because the extremum for power maximisation is not known a priori, whereas conventional model-predictive control is concerned with driving an objective function to zero, such as for tracking a power reference (Grüne and Pannek, 2017).

The full control optimisation problem is solved in a receding horizon control scheme, in which $N_c \geq 1$ is defined to be the number of samples executed before re-optimisation. Larger values reduce the computational requirements, but reduce flexibility under changing predictions as the control signal is re-optimised less frequently. At every re-optimisation step, information is shared between turbines in the farm.

The yaw reference for each individual turbine is defined by the coefficients of the spline basis, of which several are fixed and the $n_m$ free coefficients gathered in the control vector $\boldsymbol{m} \in \mathbb{R}^{n_m}$. For every turbine, we construct the scalar objective function $J : \mathbb{R}^{n_m} \to \mathbb{R}$ to optimise the mean power production over the prediction horizon for the current turbine and its immediate downstream neighbours

$$J(\boldsymbol{m}) = \sum_{k=k_0}^{k_0+N_h} \left( R(\psi_k - \psi_{k-1})^2 + \sum_{i=1}^{n_{t,sub}} Q p_{k,i} \right). \tag{19}$$

The objective function uses an initial condition $\boldsymbol{q}_{k_0}$ for the wake model at the current time step $k = k_0$ with the state update according to (1) using a set of free-stream velocity predictions $\boldsymbol{u}_\infty$ over the horizon. The power $p$ of turbine $i$ at time step $k$ is calculated following (11) and the yaw heading reference $\psi$ following (18). The output weight $Q < 0$ such that minimisation of the objective maximises mean power

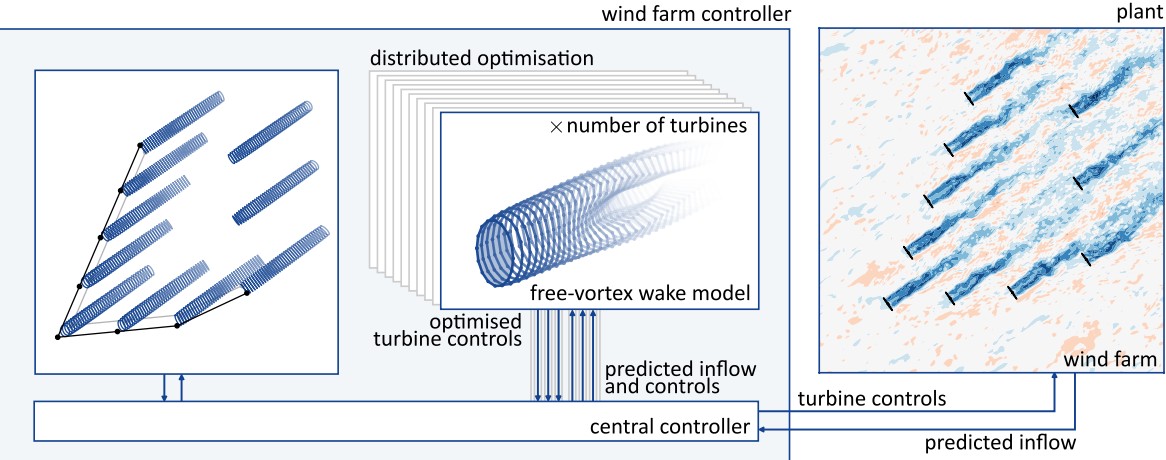

**Figure 6.** The finite-horizon optimisation problem for economic model-predictive control is solved in parallel with a wake model for each turbine. A central controller communicates with the wind farm to update control set-points and incorporates the predicted inflow. It updates the graphs connecting upstream and downstream neighbours and distributes information. The current control framework is open loop and does not utilise wind farm measurements for state estimation or parameter updates.

production over the horizon and the input weight $R \geq 0$ balances the output and actuation cost. The number of turbines $n_{\text{t,sub}}$ is size of the subset of the wind farm consisting of the current turbine and its immediate downstream neighbours in the directed graph.

The objective function is then implemented in the optimisation problem

$$\min_{\boldsymbol{m}} J(\boldsymbol{m}) \quad \text{subject to } |\gamma_k| \leq \gamma_{\max}, \tag{20}$$

where the maximum yaw offsets $\gamma_{\max}$ are enforced as hard limits relative to the predicted inflow. The optimised yaw signal does not include the induced yaw effects, these are taken into account before sending the control signals to the wind turbine yaw controller for implementation in the wind farm. The problem is solved with the BFGS optimisation algorithm (Byrd et al., 1995) although this approach was ineffective in previous work (van den Broek et al., 2022a) because of the noisy optimisation landscape. The smoothing effect of the B-spline basis enabled better convergence trial optimisations.

The controller framework presented here is operated in open loop as data assimilation for state estimation and parameter updates are beyond the scope of the current work. Additionally, management of fatigue loading is important for turbine operation, but left out of the control objective. Minimisation of fatigue loads could be achieved by integrating a surrogate model for turbine loads (Shaler et al., 2022; Bossanyi, 2022), adding an associated cost to the objective function, and subsequently appropriately balancing objective weights.

---

**Algorithm 1** Free-vortex wake controller

---

**initialise** wind farm from configuration
**for** turbine in wind farm **do**
    **construct** free-vortex wake model
    $\boldsymbol{q}_0 \leftarrow$ **run** transient with initial inflow
**end for**
$k \leftarrow 0$
$k_{\text{final}} \leftarrow t_{\text{final}}/\Delta t$
**while** $k < k_{\text{final}}$ **do**
    $\boldsymbol{u}_\infty \leftarrow$ inflow over prediction horizon
    graphs $\leftarrow$ **update** graphs with $\boldsymbol{u}_\infty$
    **for** turbine in wind farm **do**
        position, controls $\leftarrow$ downstream neighbours from graphs
        $\boldsymbol{m} \leftarrow$ **minimise** $J(\boldsymbol{m})$ with $\boldsymbol{u}_\infty$, position, controls
        $\boldsymbol{c} \leftarrow$ **combine** fixed and optimised coefficients
        $\psi^\star \leftarrow$ **spline** with coefficients $\boldsymbol{c}$
        **for** $i$ in 1 to $N_c$ **do**
            $\boldsymbol{q}_{k+i} \leftarrow$ **update** model state $\boldsymbol{q}_k$ with $\psi^\star$, $\boldsymbol{u}_\infty$
            $\gamma_{\text{ind}} \leftarrow$ **calculate** induced yaw at position
        **end for**
        $\gamma_{\text{ind}} \leftarrow$ upstream neighbours ($\leq 2$) from graphs
        $\gamma_{\text{ref}} \leftarrow$ **reduce** $\gamma^\star$ with $\gamma_{\text{ind}}$
    **end for**
    $k \leftarrow k + N_c$
**end while**

---

## 4   Methods for controller validation

Given the novel control strategy constructed around the FVW model, it is imperative to validate its control performance with a suitable reference controller and realistic operating conditions. Sect. 4.1 describes the turbine yaw controller used to implement the reference signals from the wind farm controllers. The reference wind farm controllers are introduced in Sect. 4.2, followed by the settings for the FVW controller in Sect. 4.3. The wind farms for the test cases are defined in Sect. 4.4 and a realistic time-varying wind signal for driving the simulation study is provided in Sect. 4.5. Finally, Sect. 4.6 describes the simulation environment that is used to measure controller performance.

### 4.1   Turbine yaw controller

The first aspect of testing the control strategy in a realistic wind farm setting is the implementation of a local turbine yaw controller. This yaw controller is used for all control strategies to follow the specified reference signal. The basic yaw controller is implemented based on a dead-band control strategy (Kanev, 2020) with an $8°$ dead band. When the magnitude of the yaw error exceeds the dead band, the turbine will yaw with a constant $0.3°.\text{s}^{-1}$ yaw rate until the error reaches zero. Additionally, to avoid persistent unintentional yaw misalignment, error integration is implemented similar to Kragh and Fleming (2012). The turbine will yaw until the error reaches zero if the cumulative error exceeds the equivalent of five degrees of misalignment for five minutes. This is set more strict than in the original work to facilitate a fair comparison of the control strategies.

### 4.2   Reference wind farm controllers

The standard baseline control strategy for wind farm control is greedy control, where each turbine operates individually to track the inflow wind direction without considering collective wind farm performance. This baseline is used in the current study to provide normalised output measures and quantify relative gains. However, a reference wake steering controller is necessary to assess the potential for dynamic model-predictive control.

The current industry state-of-the-art for implementing wake steering uses look-up tables with yaw angles optimised using steady-state engineering models. Therefore, we use FLORIS (NREL, 2022) with the cumulative curl model (Martínez-Tossas et al., 2019) and the serial-refine optimisation strategy (Fleming et al., 2022) to generate a look-up table with yaw angle offsets optimised for power production in steady-state. A $2°$ hysteresis is applied on the wind direction signal to avoid excessive yaw actuation around wind directions where the yaw offset in the look-up table changes sign (Kanev, 2020)

The model-predictive controller assumes a preview of the inflow over the optimisation horizon. For fair comparison, the greedy controller and the LUT controller use the same inflow information. However, these controllers lack preview and therefore utilise only the instantaneous flow conditions. Recent work by Simley et al. (2021b) and Sengers et al. (2023) explores LUT control with preview of the wind direction, selecting yaw offsets from the look-up table based on the inflow direction at a time in the future. With these studies in mind, we implement a preview-enabled look-up table (PLUT) controller to study whether results similar to the economic model-predictive controller might be realised by utilising a simple control strategy. To do so, we use the same pre-optimised yaw offsets and hysteresis strategy that the LUT controller is based on. However, the yaw reference is selected based on the inflow direction $\theta$ at a time $t_{\text{preview}} = t + \Delta t$.

We relate the preview time to the time it takes for the effects of control actions to propagate to downstream turbines. A simple formulation relates the preview window $\Delta t$ to turbine spacing $\Delta x$ and the free-stream wind speed $u_\infty$ as

$$\Delta t = \frac{\Delta x}{f_\text{w} \cdot u_\infty} , \tag{21}$$

where $f_\text{w} \leq 1$ is an approximate fraction of the free-stream wind speed at which the wake propagates. Simley et al. (2021b) and Sengers et al. (2023) find an optimal preview window which, for their configuration, is equivalent to $f_\text{w} = 0.9$ and $f_\text{w} = 1.0$, respectively. For now, we implement the control strategy with $f_\text{w} = 1$ and an inter-turbine spacing of $\Delta x/D = 5$ which corresponds with the spacing along the main rows of wind turbines where wake steering will be applied for the layouts presented in Sect. 4.4. Note that this a rough preview implementation; further exploration and refinement is outside the scope of the current work.

### 4.3   FVW controller settings

In the current study, the optimisation problem at the core of the FVW controller is solved over a prediction horizon of $N_\text{h} = 80$ steps. In order to save some computational expense, the first $N_\text{c} = 5$ samples of the optimised control signal are executed before re-optimisation, which is the first $6\%$ of the prediction horizon. The output weight is set to $Q = -1$ and the input weight $R = 0.001$ balances the output and actuation cost. The optimisation parameters were chosen based on results of exploratory parameter variations. The yaw offset results from the optimisation are limited to maximum yaw offsets $\gamma_{\max} = 30°$.

A B-spline basis with seven coefficients is chosen to provide enough degrees of freedom for control on the given prediction horizon, which corresponds to the example illustrated in Fig. 5. The first coefficient is chosen equal to the current yaw angle at time step $k_0$ to ensure a continuous yaw signal, $c_1 = \psi_{k_0}$, and the final three coefficients, $c_5, c_6, c_7$, are chosen equal to the predicted wind direction at the associated

time steps. The middle three coefficients remain free as the control parameters for the optimisation problem and are used to define the control vector $\boldsymbol{m} = [c_2 \, c_3 \, c_4]^\mathrm{T}$.

The directed graph network is constructed using a spreading angle of $30°$ and a range of $8\,D$ for both the upstream and downstream connections. The $8\,D$ range is the limit for consistent power predictions with the current settings of the FVW model because a finite-length wake is simulated.

## 4.4 Wind farm definitions

The test wind farms use the DTU 10MW reference turbine (Bak et al., 2012) with a rotor diameter $D = 178.3\,\mathrm{m}$ and a hub height of $119\,\mathrm{m}$.

The first test case is a three-turbine wind farm (TTWF), illustrated in Fig. 7, is a relatively simple proof-of-concept to test the novel control strategy under a synthetic time-varying wind direction. The turbines are aligned with a $240°$ wind direction and spaced $5\,D$ apart. The case provides room for transitions between greedy control and wake steering. It also requires the controller to account for secondary steering effects to avoid excessive yaw misalignment.

The second wind farm test case is a subset of the Hollandse Kust Noord (HKN) wind farm, scaled by rotor diameter from the actual turbine to the DTU 10MW reference turbine. The ten turbines in the South-West corner are selected as illustrated in Fig. 7. For the first HKN test case, labelled HKNA, a synthetic wind direction signal is constructed to test controller performance for several transients and steady-state wind directions. The wind direction signals for the TTWF and HKNA cases are designed specifically to test the controller performance in the respective wind farm layouts.

## 4.5 Real-world wind signal

In order to set up realistic wind variations for the wind farm, we make use of publicly available wind measurements. The raw data from a ZephIR $300\,\mathrm{m}$ wind lidar at the HKN location is adapted from the KNMI Data Platform (KNMI, 2023).

Two seven-hour time series of wind speed and wind direction are selected from the available measurements and illustrated in Fig. 8. These time series drive the LES for test cases HKNB and HKNC. The selected data have wind directions $180° \leq \theta \leq 270°$ such that the South-West inflow boundaries can be used for driving the LES domain. Furthermore, the wind speeds are such that the wind turbines operate in region II, below-rated conditions. The measurements record wind conditions at $133\,\mathrm{m}$ above sea-level, which is close to the $119\,\mathrm{m}$ hub height of the DTU 10MW reference turbine.

The raw data is cleaned up and interpolated from the original samples at approximately $17\,\mathrm{s}$ intervals to $1\,\mathrm{s}$ samples with cubic splines. A low-pass filter with a [parse-numbers = false]1/600Hz cut-off frequency is applied to generate a suitable signal for driving the LES. Higher frequency variations

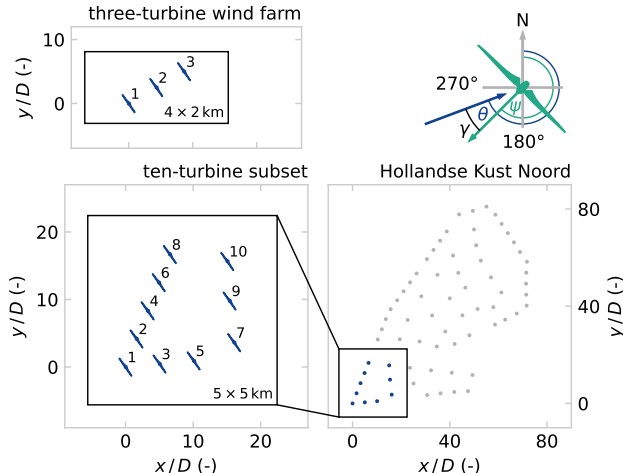

**Figure 7.** Layout of the wind farm test cases and simulation domains, and angle definitions for wind direction $\theta$, turbine heading $\psi$, and yaw misalignment $\gamma = \theta - \psi$. The three-turbine wind farm has a $5\,D$ spacing and is aligned along $\theta = 240°$. The ten-turbine subset of Hollandse Kust Noord (HKN) is the South-West corner of the wind farm, scaled to the 10MW reference turbine.

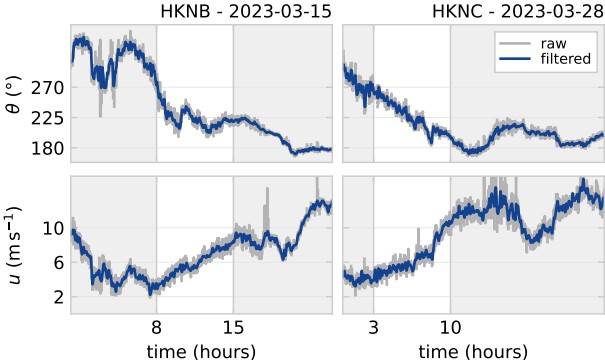

**Figure 8.** Time series of lidar measurements of wind direction and wind speed at the HKN location (KNMI, 2023). The raw data is post-processed and low-pass filtered with a [parse-numbers = false]1/600Hz cut-off frequency. Two seven-hour time series with wind direction $180° \leq \theta \leq 270°$ and below-rated wind speeds are selected for driving the realistic wind variations in the LES.

are naturally reintroduced in the turbulent variations of the simulation.

## 4.6  Simulation environment

The controllers are tested in large-eddy simulations (LES) with turbulent precursors using SOWFA (Churchfield et al., 2012). Turbines are modelled with a rotating actuator-disc model of the DTU 10MW reference turbine (Bak et al., 2012).

The three-turbine wind farm is simulated in a $4km \times 2km \times 1km$ domain. The HKN cases are run in a $5km \times 5km \times 1km$ domain. The positioning of the turbines in the domains is illustrated in Fig. 7. The base cell size is set to 20 m in all directions. A single refinement is applied to the bottom layer ($z < 300\,$m) to 10 m cells. This yields a total of approximately $9.7 \times 10^6$ grid cells. The simulations are run with a 0.5 s time step.

Turbulent precursors are prepared before the controller simulations by simulating for 20000 s to develop turbulence and then forcing the specified wind direction and wind speed variations. The wind direction and speed appear to change almost uniformly throughout the flow field.

The use of the same precursor data for all control strategies allows a comparison of the differences in output measures originating from control.

## 5  Results and discussion

The performance of the novel FVW controller is first evaluated on the three-turbine wind farm in Sect. 5.1. Subsequently, it is tested on the ten-turbine subset of HKN with synthetic wind direction variation in Sect. 5.2 and with realistic wind variations in Sect. 5.3. Section 5.4 comments on the limitations of optimisation with finite-length wakes on a finite horizon and Sect. 5.5 provides a perspective towards closed-loop control. A benchmark of computational performance is presented in Sect. 5.6 to discuss the steps towards real-time optimisation. Finally, Sect. 5.7 discusses the potential for preview-enabled look-up table control.

### 5.1  Three-turbine wind farm

The three-turbine test case is a relatively simple proof-of-concept to test the novel control strategy. The yaw offsets implemented by the three controllers are illustrated in Fig. 9. Intentional yaw misalignment is applied to turbines 1 and 2 in all control strategies. The maximum offsets utilise the $\pm 30°$ bounds applied to the optimisation problem. No offsets are applied to turbine 3, which is the most downstream turbine. It is always controlled towards alignment with the local free-stream wind direction for the range of wind directions studied here. The magnitude of yaw misalignment on turbine 2 is lower than on turbine 1 for both FVW and LUT controllers. This is the result of accommodating for secondary steering

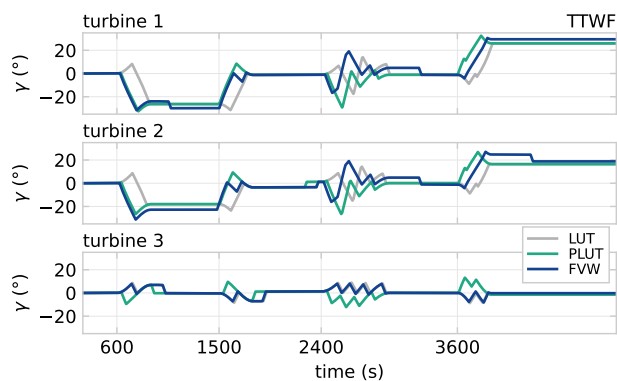

**Figure 9.** Yaw offsets realised for the three-turbine test case. Turbines 1 and 2 implement intentional yaw misalignment for wake steering around turbine 3. The FVW controller anticipates wind direction changes and accounts for secondary steering effects.

effects in the yaw control strategy. The induced yaw effect from operating in the wake of the yaw-misaligned turbine 1 lowers the required angle of misalignment for a similar level of wake redirection.

An important feature of the yaw reference generated by the novel FVW controller is the anticipation of changes in wind direction – the turbines yaw before the wind has actually rotated. The LUT controller, on the other hand, reacts to changes as they happen. The basic PLUT implementation realises an effect on the yaw reference for turbines 1 and 2 that is similar to the FVW controller behaviour by anticipating the transients. However, turbine 3, which is most downstream, tracks the instantaneous wind direction in the FVW controller, but yaws in advance of the transients with the PLUT approach. This leads to a longer time spent in misaligned operation, where yaw-aligned operation would be optimal.

The gains in power production of the FVW controller over the LUT appear mainly during

the transients in wind direction as illustrated in Fig. 10, with the PLUT controller achieving similar results. The power lost due to misaligned operation is initially sacrificed as the controller anticipates changes, which results in a gain in production following the transient. The FVW controller makes use of the dynamics of propagation of the wakes for long-term gains in power production, which can be seen in the normalised energy $E$ produced since the start of the simulation. The power gains with FVW and PLUT controllers highlight the importance of considering the wake propagation dynamics when dealing with time-varying inflow conditions. The optimisation over future inflow conditions with the FVW, as well as the inclusion of preview in the LUT can both produce control signals that provide better performance than the LUT based on steady-state assumptions. The performance in steady-state is approximately equivalent between the three wake steering controllers.

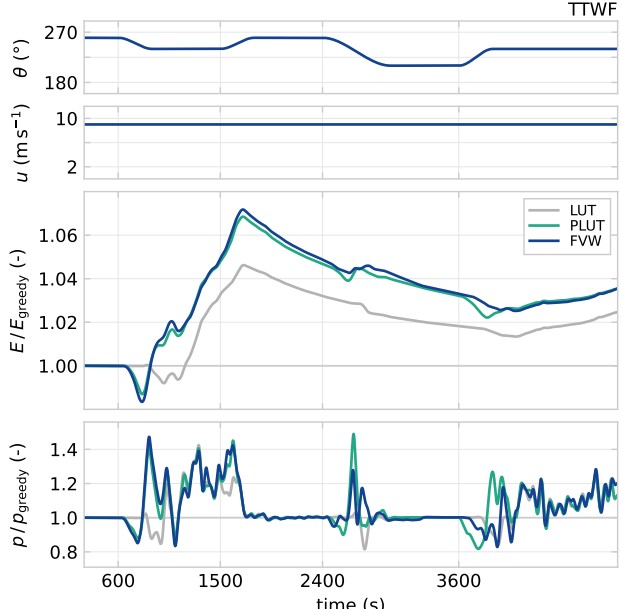

**Figure 10.** Wind farm performance for the three-turbine test case, where the top two plots show the driving wind direction and wind speed for the simulation. The third row shows energy produced relative to greedy control and the bottom row show 60 s-averages of relative power production. The FVW controller improved power generation during and following transients by anticipating changes and performs approximately equivalent to the LUT in steady state.

The cumulative results for the TTWF are shown in Fig. 11 and listed in Table 2. In terms of power production with respect to greedy control, the implementations of wake redirection with the FVW and the PLUT controllers yield a 3.8 % gain which exceeds the 2.7 % achieved with the LUT approach. The demand on the yaw actuators is measured using the yaw travel $\Delta\psi$, which is the total angular distance covered during the length of the simulation. The power improvements with the FVW are achieved with only a slightly increased demand on the yaw actuators as the total yaw travel increase compared to the greedy baseline is 58.1 % for the LUT and 69.3 % for the FVW controller. The yaw travel for the LUT and PLUT controllers is identical as they are based on the same wind direction signal and yaw offsets.

## 5.2 Ten-turbine subset of HKN

We expand the results from the three-turbine case by considering the ten-turbine subset of the South-West corner of the HKN wind farm with a synthetic wind direction variation defined in Fig. 15. A series of flow snapshots from the LES are provided in Fig. 12 to illustrate the discussion of controller performance.

The cumulative performance of the FVW with respect to the LUT and PLUT is illustrated in Fig. 13 and listed in Ta-

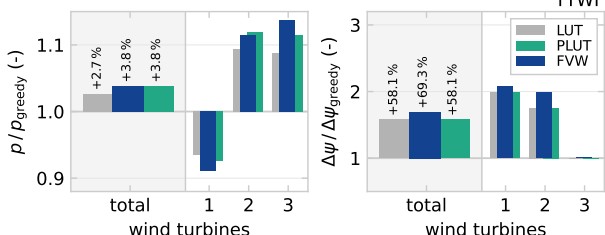

**Figure 11.** Cumulative results for the three-turbine test case in terms of total relative power production and yaw travel. The FVW controller improved power production at a slight increase in yaw travel compared to the LUT controller. In this case, the PLUT achieves the same improvement in power production as the FVW.

**Table 2.** Mean power production and cumulative yaw travel for the four test cases, where HKNA, HKNB, and HKNC feature the same ten-turbine wind farm and TTWF features a three-turbine wind farm. Increases are noted relative to the greedy control baseline.

|      |        | power (MW) |         | yaw travel (°) |          |
|------|--------|-----------|---------|---------------|----------|
| HKNA | greedy | 61.23     |         | 1247          |          |
|      | LUT    | +1.51     | +2.5%   | +691          | +55.4%   |
|      | PLUT   | +1.59     | +2.6%   | +691          | +55.4%   |
|      | FVW    | +1.96     | +3.2%   | +518          | +41.5%   |
| HKNB | greedy | 22.79     |         | 1749          |          |
|      | LUT    | +0.49     | +2.2%   | +2928         | +167.4%  |
|      | PLUT   | +0.59     | +2.6%   | +2788         | +159.4%  |
|      | FVW    | +0.53     | +2.3%   | +615          | +35.1%   |
| HKNC | greedy | 27.86     |         | 2052          |          |
|      | LUT    | +1.46     | +5.2%   | +4232         | +206.3%  |
|      | PLUT   | +1.85     | +6.6%   | +4237         | +206.5%  |
|      | FVW    | +1.58     | +5.7%   | +2780         | +135.5%  |
| TTWF | greedy | 16.36     |         | 364           |          |
|      | LUT    | +0.44     | +2.7%   | +211          | +58.1%   |
|      | PLUT   | +0.63     | +3.8%   | +211          | +58.1%   |
|      | FVW    | +0.63     | +3.8%   | +252          | +69.3%   |

ble 2. The FVW controller produces a 3.2 % gain in mean power production which exceeds the gain of 2.5 % from the LUT controller. This gain is consistent with the improvement over the LUT controller found in the TTWF case. The PLUT controller only realises a 2.6 % power gain, which is a slight improvement over the LUT, but much less than is achieved with the FVW. The FVW notably reduces the yaw travel demand, increasing 41.5 % over greedy control, whereas the LUT and PLUT controllers lead to a 55.4 % increase. This contrasts the results from the TTWF case where a slight increase in yaw travel was observed.

Turbine 1, which is upstream in all simulated wind directions, loses a bit more power comparing the FVW to the LUT as it operates under yaw-misaligned conditions for longer. However, this is offset by the power gain coming mostly

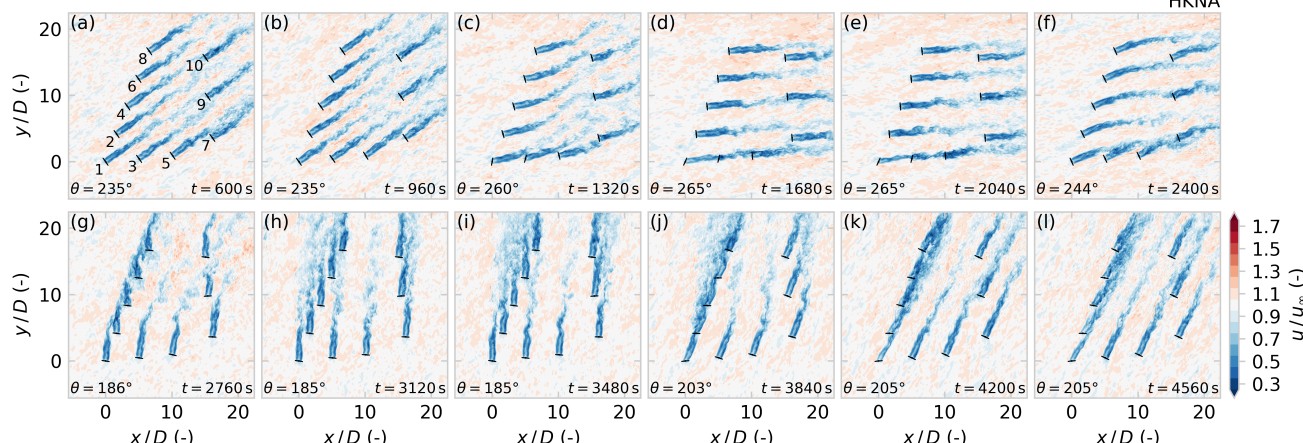

**Figure 12.** Series of hub height flow snapshots from LES of the HKNA test case with the FVW controller. In the initial transient, (a) all wind turbines are aligned with the mean inflow direction. Wake steering solutions are illustrated in (d) and (e) for the southern row of wind turbines $1-3-5$ and (k) and (l) for the western row of wind turbines $1-2-4-6-8$. Waked turbines have a reduced yaw offset because of the modelling of secondary steering effects. For certain wind directions, long wakes impact farm performance which are not accounted for in the FVW due to the limited prediction horizon. For example, (d) and (e) show the wake from turbine 4 impinging on turbine 9 and (h) and (i) show turbine 8 operating in the wake from turbine 3.

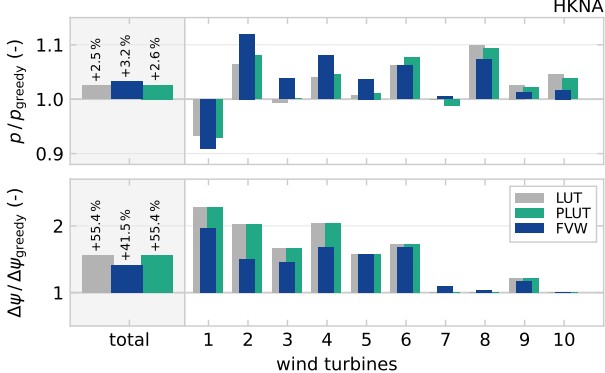

**Figure 13.** The FVW controller improves power production with respect to the LUT and PLUT approach and reduces the increase in total yaw actuation required. The gain comes mostly from turbines 2 to 7, whereas turbines 1 and 8 to 10 lose some power with respect to the LUT and PLUT controllers.

from turbines 2 to 7, which are relatively close together along the wind directions considered. Unlike the TTWF case, the power gains form the PLUT controller are not equivalent to that of the FVW controller.

Turbines 8 to 10 are further downstream and are therefore not always accounted for in the optimisation with the FVW as, for the wind directions considered, they are often beyond the finite length of the simulated wakes given the current controller settings. The implementation of preview on these downstream turbines leads to a slight loss in performance comparing the PLUT to the LUT controller.

This lack of wake redirection away from far downstream turbines is also apparent in the yaw offsets applied as illustrated in Fig. 14. Turbines 1, 3, and 4 have steady-state segments where no yaw misalignment is applied in the FVW controller, even though the LUT prescribes offsets for these wind directions. Their downstream neighbours are beyond the length of the simulated wakes with the FVW and can therefore not be accounted for in the model-predictive control optimisation with the current controller configuration.

The power generation over time for this test case is illustrated in Fig. 15. The underperformance of the FVW controller in the initial segment is due to the lack of yaw misalignment on turbines 1 and 4, which leads their wake to impinge on turbines 9 and 10, whereas the yaw misalignment specified by the LUT controller minimises this negative aerodynamic interaction. The final segment of the simulation shows particular benefit from wake steering as the wind direction is aligned with the western row of turbines 1–2–4–6–8. The gains in power for the FVW controller over the LUT controller emerge during the transients in wind direction. Accounting for the propagation dynamics of the wakes leads to fewer instances of loss compared to greedy control. In steady state, the FVW controller with the current controller settings performs approximately equivalent to, or slightly worse than the LUT controller.

The PLUT controller notably underperforms even with respect to the LUT controller for a large part of this simulation. The simple preview implementation produces some gains following the transients, but sacrifices more power to achieve this. These losses may be due to the large wind direction variations and the wind farm layout, in addition to

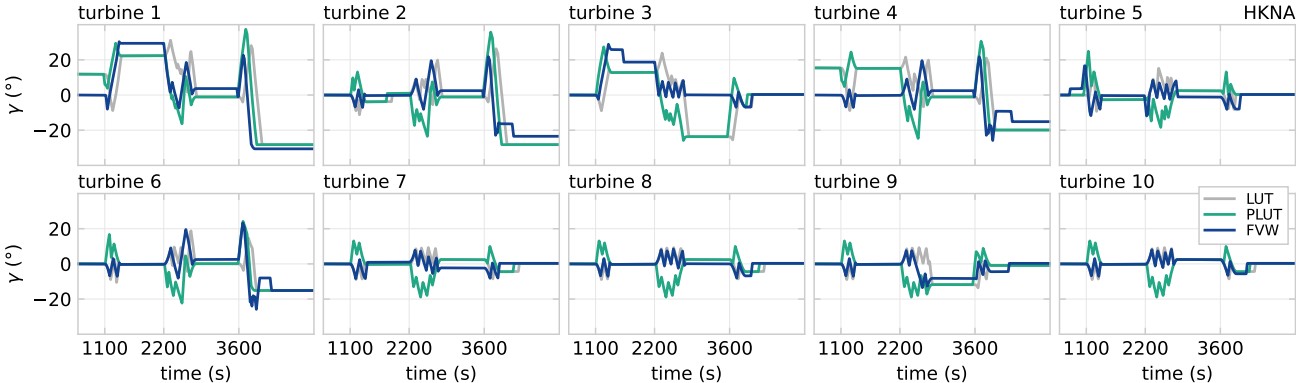

**Figure 14.** Realised yaw offsets for LUT and FVW controllers for the HKNA test case. Notably, there are some steady-state segments where turbines 1, 3, and 4 are not misaligned by the FVW controller where the LUT does prescribe a yaw offset. This is due to the limitations of the simulated wake length and prediction horizon in the current settings of the FVW controller.

the implementation of preview on turbines that should be in yaw-aligned operation.

### 5.3 Realistic wind variations

The previous two cases highlighted the potential for the gains in terms of power generation and yaw travel reductions that may be achieved with the FVW controller. The wind direction variations were, however, specifically designed to test the added value of the dynamic model-predictive control framework and therefore lack realism. The two cases HKNB and HKNC are simulated using measured wind data to demonstrate controller performance under real variations in wind speed and direction.

Figure 16 summarises the total improvement in power production with respect to greedy control, which is also listed in Table 2. In case HKNB, the increase in power generation by wake redirection is improved from 2.2 % with the LUT to 2.3 % with the FVW controller and 2.6 % with the PLUT. The increased yaw travel is limited to only 35.1 % with the FVW compared to the 167.4 % with the LUT approach. The minor differences in yaw travel between LUT and PLUT controllers are due to the treatment of the end of the time-series simulation. Case HKNC shows an increase of power production of 5.7 % with the FVW compared to 5.2 % with the LUT and 6.6 % with the PLUT, as well as a reduction of additional yaw travel from 206.3 % to 135.5 %. The losses of the FVW with respect to the LUT controller appear on turbines 9 and 10, which are far downstream from their upstream neighbours, beyond $12\,D$ downstream for most of the simulated wind directions.

These results show that some of the improvement in wind farm performance from a dynamic economic model-predictive control approach is maintained under realistic, time-varying wind conditions, where both wind direction and speed change over time. However, under certain conditions, unnecessary losses are incurred with respect to the

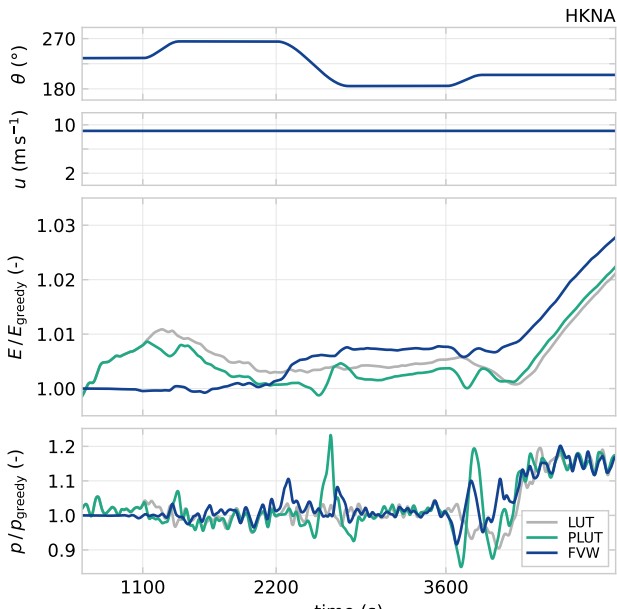

**Figure 15.** Relative energy produced and power production for the HKNA test case. During transients in wind direction, the LUT approach loses power with respect to greedy control. The FVW controller loses a bit as it anticipates changes, but then gains power over the LUT controller. The initial steady-state segment also shows underperformance with respect to the LUT approach. The PLUT controller appears less effective during transients and only slightly improves over the LUT approach.

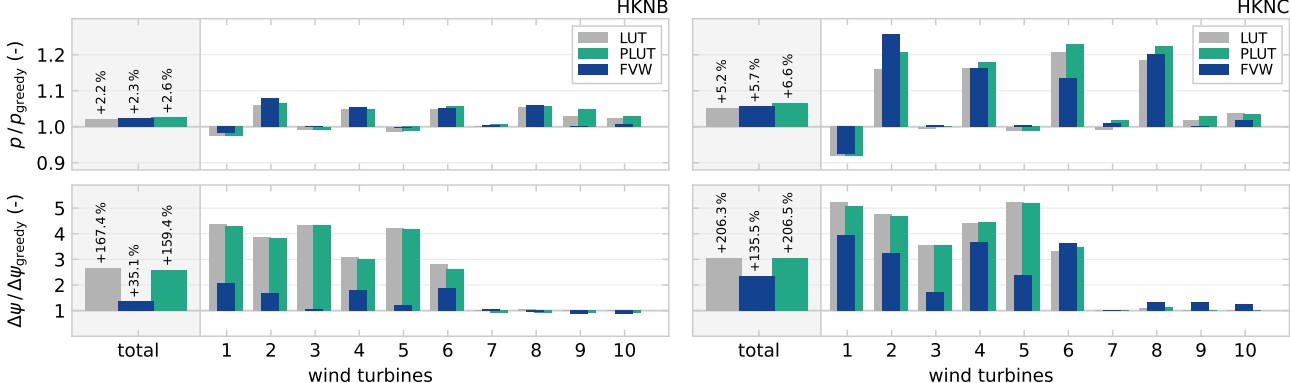

**Figure 16.** Controller performance relative to greedy control in terms of power production. In both cases, the PLUT controller yields the biggest increase in power production compared to greedy control, although the FVW controller also outperforms the LUT approach. Additionally, the FVW achieves these gains with a lower total cost in terms of yaw actuator duty. The FVW controller shows a tendency to underperform compared to the LUT for turbines 9, and 10 which are more than $12\,D$ away from their upstream neighbours for most of the simulated wind directions.

LUT controller due to the limitations of the FVW controller with the current settings. A simple preview implementation appears more effective in accounting for the effects of wake propagation, while not being limited by finite wake length simulation and receding horizon predictions.

The performance over time is shown in Fig. 17. The relative energy production over time shows that the power gains from the FVW controller over the LUT controller are consistent throughout most of the simulated time series. The performance of the FVW and PLUT controller is equivalent for large parts of the simulation. The energy production with the PLUT sometimes exceeds the FVW, but some additional losses are incurred that bring it back to the same level.

The final segment of the HKNC test case exhibits a pattern with some large performance differences between the LUT and FVW controllers. This is where the PLUT controller achieves a large gain with respect to the FVW controller, whereas they realised similar production until that point in time. This segment is illustrated in more detail in Fig. 18 with relative power production and the yaw heading of turbine 1. The wind direction oscillates slightly around $\theta = 201.5°$, which is aligned with the western row of turbines $1 - 2 - 4 - 6 - 8$. The yaw action of turbine 1 is representative of the control signal applied to turbines 2, 4, and 6 further downstream.

Due to the limits of the prediction horizon in the FVW controller, the FVW controller produces a control signal that switches the direction of wake steering with the oscillations in the wind direction. On the contrary, the implementation of hysteresis in the LUT controllers produces a consistent yaw offset reference to one side when combined with the local turbine yaw controller. Without hysteresis, the LUT controllers would present the same switching behaviour currently observed in the FVW controller.

This difference in control signal leads to significant variations in relative power production. For this wind direction variation from approximately $t = 22000\,\mathrm{s}$ to $23700\,\mathrm{s}$, the predictive action of the FVW controller anticipates gains that are not fully realised. The losses from the yaw movements exceed the gains from the wake steering in the optimal direction. The final segment from $t = 23700\,\mathrm{s}$ onwards shows how the predictive controller anticipates the wind direction variation to yield a net gain in power production compared to the LUT. The PLUT controller is able to realise these gains without the losses incurred with the FVW control signal and ends up with the largest average power production.

## 5.4 On wake length and the prediction horizon

The results from the control test cases show some of the limitations of the proposed model-predictive control approach. The finite-horizon optimisation can not account for turbines that are outside the simulated wake length or beyond the prediction horizon.

If wind turbines are placed along a straight line, the simulated wake and optimisation horizon only needs to be long enough to cover optimisation from one turbine to the next downstream neighbour to trigger wake steering. However, for longer rows of turbines, the segment from HKNC shown in Fig. 18 demonstrates that longer horizons will probably be beneficial to avoid excessive switching of the wake steering direction.

For large inter-turbine spacing without intermediate downstream turbines, long wakes will need to be simulated with long prediction horizons to be able to properly account for the downstream effects and reach wake steering yaw control solutions. This limitation is apparent in the lack of performance improvement for turbines 9 and 10 in all the HKN cases. For most of the wind directions under consideration,

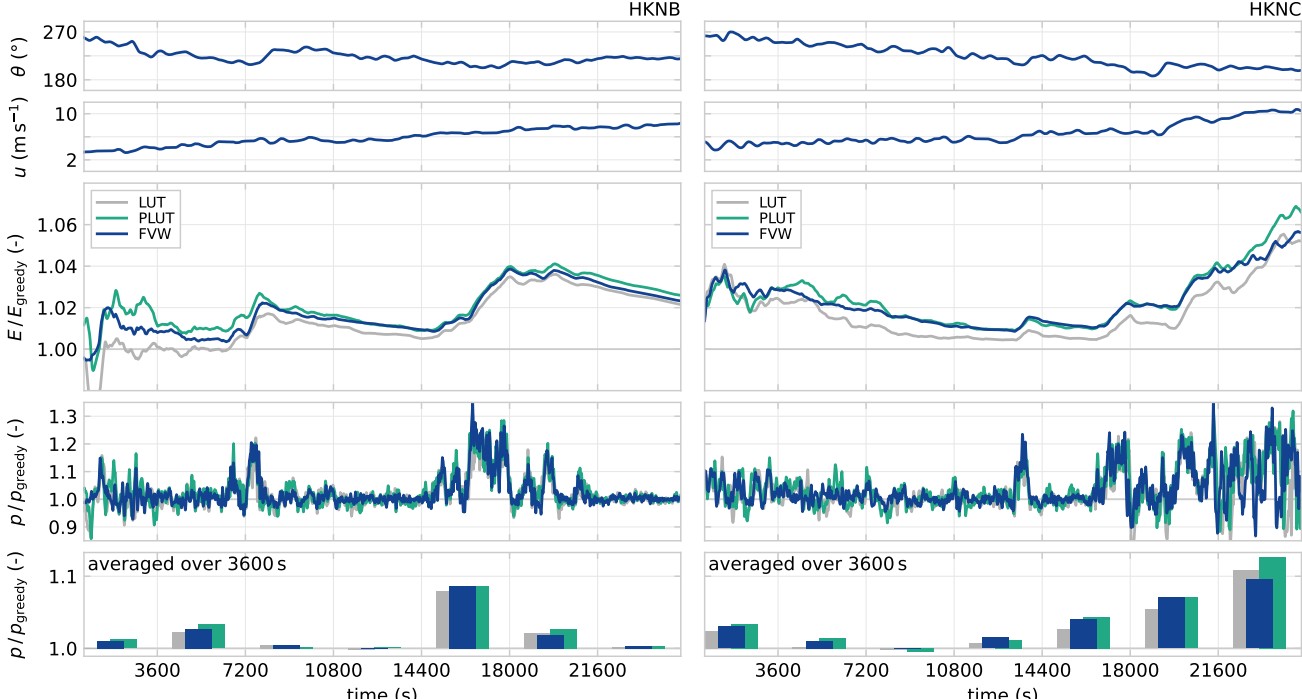

**Figure 17.** Relative energy produced, power over time, and power averaged in 3600 s bins for the two data-driven test cases, HKNB and HKNC. The driving wind direction and wind speed are shown in the top two rows. The third row shows the cumulative energy time series normalised with respect to the greedy baseline controller. Both the LUT, PLUT, and the FVW controller exhibit significant improvements over greedy control. The bottom row shows the one-hour averaged power production of both controllers normalised by the greedy baseline.

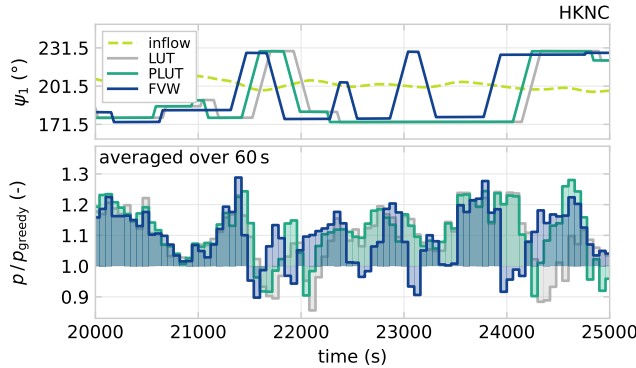

**Figure 18.** Segment of the HKNC test case shown in Fig. 17. The inflow wind direction oscillates around the western row of turbines $1 - 2 - 4 - 6 - 8$, which is aligned at $\theta = 201.5°$. The yaw heading of turbine 1, $\psi_1$, is representative of that applied to turbines 2, 4, and 6. The FVW controller switches wake steering directions from $t = 22000$ s to 23700 s, whereas the hysteresis in the LUT controller produces a constant yaw offset. The excessive yaw action in the FVW results in underperformance for this segment. Beyond $t = 23700$ s, the FVW correctly anticipates the wind direction variation producing a net gain in power production.

they are too far downstream to be accounted for in the finite-horizon optimisation with the FVW. Very long prediction horizons would be necessary to account for downstream effects, but long prediction horizons come at considerable computational cost as both simulating longer wakes and longer prediction horizons increase computational expense. Doubling both the length of the wake and the prediction horizon would lead to roughly an eightfold increase in computation time. Maintaining a similar degree of freedom in the control signal by also doubling the number of free spline coefficients then yields an optimisation problem that is approximately $16\times$ more expensive. Additionally, longer wakes stretch the limits of what can be predicted with the physical model due to inherent instabilities in the free-vortex methods.

The steady-state optimisation with FLORIS does include these long wakes because it essentially solves a mean-flow, infinite horizon version of the control problem. For steady wind directions, the optimal yaw angles for wake steering from the steady optimisation can yield higher power production than those found through receding horizon control with finite-horizon optimisation.

Furthermore, due to the bimodal nature of wake steering, the receding horizon controller may end up implementing yaw offsets in the suboptimal direction, where the cost to switch directions may not outweigh the gain in power over the finite horizon, even though that may be optimal in an infinite-horizon sense. The steady-state optimisation does not suffer from this limitation, but will lose power when atmospheric conditions violate the mean steady-state assumptions too much. The LUT approach might then apply yaw misalignment to redirect wakes around turbines which will not propagate there due to variations in wind direction. This sacrifices power generated for an expected return that is never achieved. This is the result of a lack of inclusion of dynamic effects such as continuously varying wind conditions and propagation of wakes.

The optimal control approach might combine aspects from both receding horizon control and infinite horizon optimisation. This could enable synthesis of a controller that consistently converges to optimal solutions in steady state, while incorporating the dynamics of wake propagation for power gains during inflow transients.

## 5.5 Closing the loop

The current performance achievements are realised with an open-loop controller architecture by assuming a reasonably accurate model and wind speed and velocity predictions. However, Fig. 2 already highlights differences between the simulation framework and modelled wake deflection. It shows that the incorporation of secondary effects of wake steering is only an approximation. Furthermore, the modelled deflection is symmetric for positive and negative yaw misalignment, whereas a clear asymmetry appears in the LES data.

For adaptation of model errors and incorporation of measurements into the model state, a closed-loop control framework is required. This would allow the control strategy to adapt to varying atmospheric conditions such as veer, shear, and turbulence intensity, as well as tune model parameters such as the turbulent growth parameter $\delta$ or the yaw exponents $\beta_{\mathrm{p}}$ or $\beta_{\mathrm{t}}$.

One strategy that is promising for closing the loop is the Ensemble Kalman filter (EnKF), which has previously been developed for state estimation adaptation of steady-state models (Howland et al., 2020; Doekemeijer et al., 2020). Becker et al. (2022a) developed the EnKF for wind field estimation in a model-based setting with dynamic engineering wake model. Additionally, Shapiro et al. (2019) showed that closing the loop allows inclusion of unmodelled dynamics.

## 5.6 Towards real-time control

In order to verify the potential for real-time control, a small benchmark is run on a regular laptop running Windows 10 on

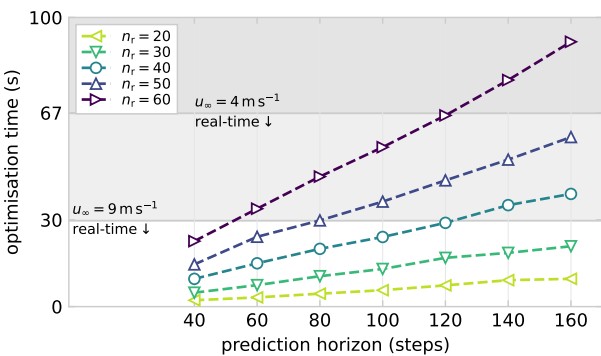

**Figure 19.** The benchmark for the computational cost of control optimisation shows how the optimisation time scales approximately linearly with the number of steps in the prediction horizon. The computational expense scales quadratically with the wake length, determined by the number of rings $n_{\mathrm{r}}$ in the wake simulation. The horizontal lines provide an indication of the level below which real-time control optimisation is achieved for $u_{\infty} = 4\,\mathrm{m.s}^{-1}$ and $u_{\infty} = 9\,\mathrm{m.s}^{-1}$ with an update every five discrete time steps.

an i7-8650 CPU at $1.90 GHz$ with $8GB$ RAM. The benchmark is run in Julia 1.8.0 using the BenchmarkTools module.

The results of the benchmark are shown in Fig. 19 to illustrate the scaling of the computational cost for solving the optimisation problem. The optimisation time scales linearly with the length of the prediction horizon and quadratically with the wake length, which is determined by the number of rings $n_{\mathrm{r}}$ in the wake simulation. The cost of the optimisation also scales linearly with the number of control degrees of freedom, which is set to three as is done throughout the current work.

The non-dimensionalisation of the FVW by rotor diameter and wind speed leads to a dependency on wind speed in measuring the performance relative to real-time. Therefore, we report values for rotor diameter $D = 178.3\,\mathrm{m}$, and relative to the inflow wind speeds $u_{\infty} = 4\,\mathrm{m.s}^{-1}$ and $u_{\infty} = 9\,\mathrm{m.s}^{-1}$. With the configuration as used in the current work, a simple forward run of the wake model with power predictions for two downstream neighbours over the full prediction horizon requires approximately 0.7 s. This means predictions can be made $1528\times$ faster than real-time at $4\,\mathrm{m.s}^{-1}$ and $679\times$ faster than real-time at $9\,\mathrm{m.s}^{-1}$. The current update rate in the model-predictive controller is fixed at every five discrete time steps; this is equivalent to an update every 67 s at $4\,\mathrm{m.s}^{-1}$ or 30 s at $9\,\mathrm{m.s}^{-1}$. With the current optimiser settings, every re-optimisation step takes about 21 s per wake, which is, respectively, $3.2\times$ or $1.4\times$ faster than real-time for optimising control updates.

This means that the current optimisation set-up realises real-time optimisation for model-predictive wind farm flow control in below-rated conditions. For that, a single processor per wake needs to be available to distribute the optimisa-

tion problems. Faster wind speeds require faster optimisation to achieve real-time model-predictive control. This might be within reach with improvements in the numerical algorithm or using a more performant processor.

## 5.7 Preview-enabled look-up table control

Under the realistic wind variations that drive the HKNB and HKNC cases, our simple preview implementation combines the effectiveness of the steady-state optimal yaw offsets with a simple strategy for accounting for wake propagation. The PLUT controller achieves a further increase in power production over the FVW controller, whereas, in the HKNA case, it underperforms significantly. The difference between these cases appears to originate from the magnitude of wind direction changes, where the FVW controller is more flexible to adapt to a broader range of circumstances.

Despite the lack of flexibility, the results demonstrate that a simple preview approach may realise power gains equal to, or greater than a more complex, economic model-predictive controller with limited simulated wake length and prediction horizon. Further refinement is required to maximise the gains that may be achieved by preview-enabled look-up table control and realise consistent performance, avoiding the losses on downstream turbines and for large magnitude wind direction variations.

Such further refinements should consider tuning the preview window to the wind farm layout, where a dependence on the wind direction would allow the preview controller to account for variable turbine spacing along different rows. Additionally, the LUT should be referenced without preview for turbines whose wake does not impinge on downstream rotors, which means that yaw-aligned operation is optimal. These adjustments are already naturally included in the economic model-predictive control optimisation, which may, therefore, provide a foundation for refining preview control.

## 6 Conclusions

A novel distributed, model-based approach to dynamic wind farm flow control is presented with a focus on yaw control for wake redirection. Previous optimisation results with the FVW are extended to economic model-predictive control at the wind-farm scale by parallelising optimisation, connecting individual models into a directed graph network, and incorporating secondary steering effects. The low computational cost enables real-time optimisation in below-rated conditions.

The novel controller is tested in a large-eddy simulation environment and compared against the industry state-of-the-art approach to wake steering, which is based on look-up tables, as well as an extension with wind direction preview. Given two wind farm configurations under synthetic wind direction variations, the FVW controller achieves improvements in power production during wind direction transients.

In the simple three-turbine wind farm, equivalent gains are achieved by the PLUT, whereas it underperforms in a ten-turbine subset of the HKN wind farm. Under realistic inflow variations, the PLUT controller yields the largest improvement in power production over the LUT. The FVW yields a smaller power gain because some undesired effects still appear in the control signal. However, in most cases, the FVW controller reduces the increased demand on yaw actuation for wake steering which is advantageous for practical application in large wind farms.

The results with the FVW and PLUT both emphasise the value of including the dynamics of wake propagation for wake steering control. Further refinements in preview-enabled control are worth investigating and perhaps insights from the model-predictive control solutions can guide the development of preview strategies for look-up table controllers.

Improvements in the FVW control strategy could be achieved by considering longer prediction horizons to accommodate wake steering for longer wakes. However, this comes at a significant computational cost for the receding horizon optimisation. The FVW dynamics are a simplified representation of reality, in this case the LES, resulting in model errors that may be minimised. For example, the inclusion of asymmetry in wake steering is also important for maximising the potential gains in wind farm power production.

Lastly, closing the loop with state feedback is an essential next step to realising dynamic yaw control in a realistic setting as it enables adaptation of model parameters to changing environmental conditions. Furthermore, the results should be extended to use realistic forecasting of future inflow conditions.

**Code and data availability.** Data and code are available at 10.4121/50138917-cf01-4780-9d1d-443593b7e974 (van den Broek, 2023).

**Author contributions.** Maarten J. van den Broek: conceptualisation, methodology, software, validation, investigation, writing – original draft, visualisation. Marcus Becker: conceptualisation, writing – review & editing, Benjamin Sanderse: writing – review & editing, supervision. Jan-Willem van Wingerden: writing – review & editing, conceptualisation, resources, funding acquisition.

**Competing interests.** At least one of the (co-)authors is a member of the editorial board of Wind Energy Science.

**Acknowledgements.** This work is part of the research programme "Robust closed-loop wake steering for large densely spaced wind farms" with project number 17512, which is (partly) financed by the Dutch Research Council (NWO).

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
