# Peer review of "Dynamic wind farm flow control using free-vortex wake models"

_Wind Energy Science, 2023_

## Author Comment (AC1)

**Response to Reviewer Comments**

First of all, we thank the reviewers for taking the time to read our manuscript and providing their and providing their positive and constructive feedback. This document provides our response and describes the changes that have been made in the light of the reviewer comments. In the subsequent sections, we will address the review reports from each of the reviewers individually.

**1 Reviewer 1**

**1.1 General comments**

> Dear authors,
> Thank you for the opportunity to review your paper. The work presents some exciting new capabilities for model-based wind farm control using free-vortex methods. The results demonstrate potential for improved wind farm performance through anticipating transients and accounting for propagation dynamics and secondary wake steering effects. I enjoyed reading through the details and believe the paper will be a strong contribution once the following points are addressed:

Dear Jaime, thank you for your positive comments and thorough review of the manuscript!

> Table 1: There have been some recent advancements in quantifying yaw control costs that indicate the parameters in Table 1 may need updating. In particular, the literature suggests $\beta_p$ may be lower than 3 (see Howland et al. (2020)) and $\beta_t$ closer to 2 in reality (see Li and Yang (2021)). Additionally, $\beta_p$ and $\beta_t$ are not constants and depend on the thrust force (See Heck et al. (2023)) and the heterogeneity of the wind field (See Liew et al. (2020)) It would strengthen the work to acknowledge these recent findings and discuss any implications on your modeled results.

The value for these coefficients may indeed be different than currently used. The values in the manuscript are based on a comparison with wind tunnel data (van den Broek et al., 2023). For completeness, we extend Section 2.1 with a note describing the possibility for these coefficients to differ in reality and be dependent on thrust force and wind field heterogeneity. They are good examples of parameters that might be refined in a (future) closed-loop implementation.

> Line 110: The assumption of optimal induction is reasonable for your below-rated cases. For completeness, considering control actions above rated conditions would further generalize the approach and should be discussed or, ideally, implemented. This could be as simple as using a $c_p - c_t$ look-up table.

We focus on below-rated cases as that is where wake steering is most valuable. To emphasise that this approach is also general enough for above-rated conditions, we now note in Section 2.1 that the induction factor – which determines thrust and power coefficients – may also be adjusted. Future work could look into extending the current framework with a simple rotor model that includes pitch and tip-speed ratio for implementing a $c_p - c_t$ look-up table.

> Figure 4: This figure shows multiple turbines with free-vortices, but you mention you are only modelling 1 vortex wake at a time. Some clarification on how you are modeling secondary wake effects with only a single wake in the free vortex model would help interpretation of the results. Are any simplifying assumptions made? How is $u_r$ determined for downstream turbines when you are only simulating a single wake in your free-vortex model?

The wakes in Figure 4 have been modelled one at a time as mentioned. The associated caption has been updated to emphasise that. The secondary steering effects are modelled based on the induced yaw angle proposed in Section 2.1, which is used to reduce the commanded yaw offset sent to the wind farm.

The rotor-averaged wind speed $u_r$ at a downstream rotor is calculated based on the free-stream velocity and velocity deficit from the wake model, which is now further specified at the end of Section 2.1. This rotor-averaged wind speed is used for power estimates in the optimisation problem, whereas the individual wakes are simulated based on the free-stream wind speed.

> Line 130: you mention a simulation study used to determine a method for induced yaw angle but do not provide details of the results of this study. Consider showing some results to justify this method.

Results from this simulation study have already been included in Figure 2, which shows a comparison of the wake deflection results from large-eddy simulation with the superposition of wakes simulated with the free-vortex wake.

> Additional details on the graph network topology selection (e.g. how are the sector angle and length determined?) would improve reproducibility and clarity of the control design.

The motivations for choosing the arc sector parameters have been provided in Section 2.3, which has now been extended to provide some more details. The radius depends on the simulated wake length; it is limited to a distance that is adequately covered by the wake for reliable predictions. The angle limits modelling connections to turbines that may interact as the flow propagates along the inflow direction. It should be at least wide enough to cover the wake width and possible deflection due to yaw misalignment. The choice of these parameters is also noted in Section 4.3.

> Section 3: Additional details on the controller iterative scheme would improve reproducibility and clarity of the control approach. In particular, what measurements are used as inputs to the controller, and how (and when) are these measurements connected to the control algorithm? Are the measurements received by the controller incrementally as new data is received? What information exactly is shared between the independent optimisations? A more concise controller definition, and perhaps a pseudocode of the algorithm, would help clarify these points.

Thank you for the suggestion; we have now included a description of the control algorithm in pseudocode as Algorithm 1 in Section 3.2. We hope this helps clarify the description of the iterative scheme. Additionally, we would like to note that for reproducibility, all code developed for this manuscript is made available as supplementary material. There are no measurements in the current control scheme as we further explain below in response to your question on closed vs. open-loop control.

> Line 198: Just out of curiosity, what do you think causes the set point at the horizon to approach greedy control? I would expect it to approach a steady-state optimal for the wind farm.

The convergence back to greedy control is a property of optimisation on a finite horizon with an economic objective. Over the prediction horizon, there exists a point when the control action at the upstream turbine no longer reaches the downstream turbine due to the delays in wake propagation. It is then optimal, in terms of total power production, for the upstream turbine to return to greedy control. This non-cooperative control action does not propagate to the downstream turbine within the horizon and therefore does not lead to losses within the finite prediction horizon. This is naturally undesirable on the infinite horizon because the wake does

actually reach the downstream turbine.

> Section 4.5 There appears to be turbulence in the LES simulations but there is no mention of turbulence in the manuscript. Furthermore, realistic low-frequency wind variations are mentioned, but again, a mention of turbulence is absent. How does the controller account for and respond to turbulence? Expanding the analysis to include effects of turbulence would provide valuable insights into robustness. Even simple benchmarking studies on a few turbulence levels could indicate where the controller succeeds or struggles.

Section 4.6 introduces the large-eddy simulation environment and notes that they are run with a turbulent precursor. It is mentioned in Section 2.1 that the wake model does not include turbulence, but the effects of turbulent and viscous diffusion are approximated using the growth of the vortex core. Further studies under varying conditions would be valuable future work, but we note that the results already presented do show that there is potential gain from incorporating wake propagation dynamics in a predictive control strategy.

> In connection with the previous point, how are measurements incorporated, and how are measurement uncertainties dealt with?

The current work assumes that a reasonably reliable prediction of the free-stream inflow over the prediction horizon is available. Measurements have not been incorporated. Uncertainty is naturally dealt with through the receding horizon control framework, which updates control signals if the inflow conditions are updates. Figure 6 and its caption are adjusted to clarify that the predicted inflow is provided to the controller instead of wind farm measurements.

> Figure 9: Why does FVW respond earlier than LUT? Is the preview you mention earlier implying that FVW receives the measurements earlier than LUT? What happens if LUT also receives (and responds to) the measurements with preview? Would this perhaps be a more fair comparison?

The FVW controller acts on a preview of the wind signal, which is used for the simulation and optimisation over the prediction horizon. On the other hand, the LUT controller acts on the instantaneous wind direction signal.

Motivated by the suggestions for preview control made by you and Michael Sinner, we have now included results with a preview-enabled look-up table controller in the manuscript. This includes a description of the control strategy in Section 4.2, an extension of Section 5 in general with additional results and discussion, further discussion in Section 5.7, and adjustments in the abstract, introduction, and conclusions. We believe this has added valuable insight and contributed to making the manuscript stronger.

> Section 5.5 The transition from formulated closed-loop to open-loop implementation seemed unclear. The formulation in Section 3 and the illustration in Figure 6 appears to describe a closed-loop, but it is revealed here in Section 5.5 that the presented work is open-loop. Some clarification on this its implications would be helpful.

We agree that the open-loop nature of the controller was presented in a manner that may have been unclear. The envisioned control framework is closed loop, but the current implementation is open loop. Figure 6 has been adjusted to specify that the controller receives an inflow prediction, but not measurements from the wind farm. Additionally, a specific note has been added to the introduction of Section 3, the body of Section 3.2, and the caption of Figure 6 to emphasise that the controller is open loop. The possibilities for closed-loop control are already discussed in Section 5.5.

> Section 5.6 The computational speed results are promising, but more details of how the method scales would be more useful to the reader to gauge what scale of problem this method is suitable for. Further benchmarks on how the method scales with problem size would indicate limits and suitable conditions for real-time application. For example, a plot showing control horizon or wake length versus computational time.

A paragraph has been added to Section 5.6 describing the scaling of computational cost. Furthermore, as suggested, Figure 19 is added to illustrate the scaling of computational time for control optimisation with the wake length and prediction horizon.

**1.2 Minor comments**

> Line 82: the subscripts 1, 2, and 3 have been used to refer to the three spatial dimensions, however you use $x_1$ and $x_2$ to refer to the start and end point of a vortex filament. To avoid confusion, perhaps use different subscripts for the start and end points, for example, $x_s$, and $x_e$.

The subscripts that denote the start and end point of the vortex filaments are chosen to maintain consistency with our prior works. Additionally, the scalar variables $x$, $y$, and $z$ have been used to refer to the three Cartesian spatial dimensions. Therefore, we believe it is appropriate to maintain the subscripts used in the manuscript.

> Line 137: change "we a method" to "we present a method"

Thanks for noticing, it has been adjusted.

> Figure 8: Please make the shaded regions more defined as they are quite faint on a black and white print out.

The figure has been adjusted to a slightly darker shade.

> Figure 10: Please define "E", and perhaps the other terms if they have not already.

The definition for the energy production $E$ is now included in Section 5.1 where Figure 10 is introduced.

> I believe these suggestions can significantly strengthen the work and hope they are received in the constructive manner intended. I look forward to seeing an updated version addressing these points as the core ideas show exciting potential.
>
> Sincerely,
>
> Jaime Liew

Thanks again for your constructive comments! We agree that they have contributed to further strengthening the manuscript.

**2 Reviewer 2**

> Good work, clearly presented. I concur with many of the comments made by Jaime Liew, and would just add a few more points:

Thank you for the further constructive comments! Our response to Jaime's comments is provided in the previous section, the response to your further comments is provided below.

> Some readers would appreciate an explanation of the term 'economic' as applied to model predictive control.

We have added the following description to the manuscript in Section 3.2: "This is an economic model-predictive control problem because the extremum for power maximisation is not known a priori, whereas conventional model-predictive control is concerned with driving an objective function to zero, such as for tracking a power reference (Grüne and Pannek, 2017)."

> Results for TTWF: FVW shows higher power gain than LUT, but with higher yaw activity too. If you make the LUT controller a bit more responsive (by reducing the dead-band for example), so that it matches the yaw travel of FVW, the power gain would presumably be improved, so this should be tested to see if the power gain remains lower than with FVW. When comparing a complicated new controller to an existing simple one, it's always important to be sure that simple tweaks to the existing controller can't do the job just as well.

The controllers are tested on a level playing field with the same turbine yaw controller. Reducing the dead-band to 1° actually has a negligible effect on the yaw travel for the LUT controller in the TTWF case. Additionally, such a tight dead-band would yield excessive yaw travel if more noise were to be present on the wind direction signal. Evaluating the TTWF case with this reduced dead-band on the turbine controller yields a mean power of 13.83 MW with the LUT controller and 17.09 MW with the FVW controller. This is an improvement of +1.5 % of the FVW over the LUT, which even exceeds the +1.1 % that is reported in the manuscript. We deem that this indicates the results as presented in the manuscript provide a fair comparison of the two controllers.

> The results for HKN do suggest an improvement in both power gain and yaw travel with FVW, which is promising, although this is clearly sensitive to the low-frequency wind variations as the benefits vary significantly between the different cases, so I would say that a lot more simulations over different randomly selected conditions, or over a much longer period of historical site data, are required in order to give confidence that the additional complexity of the proposed method is really worthwhile.

The presented results demonstrate potential gains for the inclusion of dynamics in predictive wind farm flow control as well as bringing to light some of the limitations of the proposed implementation of control optimisation with the free-vortex wake models. Doing a lot more simulations, however valuable for providing further evidence of robust performance, is outside of the limitations for the current project.

> I think it would be worth further investigation of the practicality of the proposed approach when applied to a large wind farm, both in terms of computation time, and also considering that (especially in low turbulence conditions), wakes may persist far enough downstream to affect a number of turbines, in a way which the free vortex wake method may not be able to capture accurately.

Further investigation on large wind farms is valuable for future work, but unfortunately infeasible

within the current scope. We agree that wakes may persist further downstream than is currently modelled in the free-vortex wake; evidence of that is presented in Sections 5.2 and 5.3. The inter-turbine spacing in the HKN layout is sufficiently large to exceed what can be accounted for with the FVW using the configuration in our manuscript. We further comment on the limitations of the free-vortex wake methods for simulating long wakes in Section 5.4.

In terms of computation time, the distributed approach scales linearly in the number of turbines, which means the same real-time performance can be maintained as long as the number of available processors equals the number of turbines, i.e. wake models. Further details on computational scaling have been added to Section 5.6.

> The paper makes no mention of turbine loading, although this is known to be an important issue for the practicality of wake steering. Some comments should be made to indicate how the proposed method could be extended to take loading into account.

The objective function may be extended to balance power maximisation and minimisation turbine loading. To do so, a surrogate model for turbine fatigue loads might be implemented in the modelling framework and subsequently balanced with suitable objective weights in the objective function. A description of this possibility for including loads has been added to the end of Section 3.2.

**3 Reviewer 3**

> This paper presents a method for implementing model predictive control for farm-wide power maximization. Overall, I found the paper well presented, and I particularly appreciate the detailed analysis and insights on the results obtained. I have one larger comment and a handful of smaller comments to improve the paper.

Dear Michael, thank you for the positive remarks and constructive comments on our manuscript!

**3.1 General comments**

> My main comment pertains to the use of the lookup table using the current wind conditions. The FVW received preview wind direction information, which allows it to "pre-actuate" in anticipation of wind direction changes, whereas the reference LUT approach (as well as the greedy control) receive only the current wind condition. As the authors point out, the preview gives an advantage to FVW over the LUT approach when there are changing wind conditions. In Fig. 9, it appears that the main effect is that the FVW yaw signal leads the LUT signal during changing conditions. However, if the preview of wind direction is assumed available, couldn't this also be fed to the LUT (or greedy yaw controller, for that matter) instead of the "current" wind direction to overcome the latency in the yaw system? The precise amount to advance the wind direction signal to the LUT would likely be outside of the scope of this work, but looking at Fig. 9, about 100s of advance would likely help the LUT approach significantly (probably you can find a more precise number with the data you have from Fig. 9). I would be interested to see this case added as another "reference" case, to see whether the advantages for the FVW still persist over this "preview-enabled LUT" approach.

Motivated by the suggestions for preview control made by you and Jaime Liew, we have now included results with a preview-enabled look-up table controller in the manuscript. This includes a description of the control strategy in Section 4.2, an extension of Section 5 in general with additional results and discussion, further discussion in Section 5.7, and adjustments in the abstract, introduction, and conclusions. We believe this has added valuable insight and contributed to making the manuscript stronger.

**3.2 Smaller comments**

> My smaller comments are as follows:
> 1. I feel that some of the terms used to describe the MPC problem are misleading. First, what makes this an "economic" MPC? This term is used several times, but to me, the MPC cost function (eq. 18) doesn't look especially "economic". Second, as I understand the method presented, there is no communication between the turbines. Each solves their own MPC problem without knowledge of the solutions of the other turbines. If that is correct, I would not describe this as a "distributed" MPC, as "distributed" usually implies communication between agents over a network. I'm not actually sure what the correct terminology for this set-up is (if I'm understanding it correctly); perhaps "isolated" or similar?

The distinction between conventional and economic model-predictive control is in the objective function. This terminology is from the control community, where an economic objective is one that maximises production or minimises cost without a priori knowledge of the extremum, whereas a conventional objective can be driven to zero, such as for tracking a reference. We have added this clarification to Section 3.2.

We believe that referring to the current framework as 'distributed optimisation' or a 'distributed approach' – as is done in the manuscript – is appropriate. The farm-scale optimisation problem is distributed over the individual wake models, where information is shared according to the graph networks at every receding horizon iteration. Additionally, the optimisation problems are distributed over processor cores instead of solved sequentially.

> 2. As I understand the conclusions of the paper, the key area of performance benefits for FVW over the LUT approach is in dynamic conditions. How often do these dynamic wind direction conditions actually occur, compared to steady conditions? Do they happen often enough to justify using FVW, especially given the possible underperformance (due to the finite prediction horizon length) during steady conditions?

The presented wind direction variations do not seem to be exceptional. We went through about two weeks of data to find suitable samples for the HKNB and HKNC cases, which maintained the South-West inflow directions and below-rated wind speeds. More cases with wind direction variation are present if those restrictions are loosened. Additionally, gains from dynamic control already appear in $10°$ to $20°$ shifts in wind direction, which are not particularly large. Full analysis of the advantage of dynamic control for wind farm flows is a topic for future research.

> 3. I found Section 2.1 a bit difficult to follow. Various physics-based equations are given, then a generic model form (eq. 9), followed by more physical equations. Consider reordering this section for clarity.

We have restructured the content of Section 2.1 and hope that this clarifies the brief introduction of the model that was developed in previous work.

> 4. I find the notation $u_i(x_0, x_1, x_2)$ in equation 1. confusing. I understand that $u_i$ is the velocity at point $x_0$, but it depends on $x_1$ and $x_2$ (via the $r_j$); is that correct? Although the notation "$u_i(x_0, x_1, x_2)$" may be correct, I would suggest using simply "$u_i(x_0)$", as I would find it clearer. Feel free to disagree with me here.

We agree that the suggested notation may be slightly clearer and have adjusted it.

> 5. Eq. 8—is this also based on van Kuik? Are there any assumptions made there?

Equation (8) requires no additional assumptions beyond those required for the free-vortex methods. It states that the particles are convected according to the free-stream velocity and the induced velocity from all vortex filaments. There are no other terms in the model which are neglected to come to this result.

> 6. Lines 197–202. The authors mention that $c_5$, $c_6$, and $c_7$ are matched to the wind direction, which forces greedy (i.e. suboptimal in the infinite horizon) behavior at the end of the horizon. Why is this suboptimal behavior enforced? Instead, couldn't one enforce that $c_5 = c_6 = c_7 = c_4$, so as to reduce the spline dimension while allowing the turbine to maintain the offset at $c_4$? This may also not be optimal for the infinite horizon, but it seems to me a better approximation of optimality (without haven proven anything, admittedly).

Coupling the coefficients in the way you suggest would bring forward the undesired effects from the finite horizon. The pressure towards greedy control in the later coefficients will then also affect the results at earlier coefficients. The finite horizon effects are treated in this work by making sure that the coefficients are sufficiently decoupled and the executed control action sufficiently removed from the horizon effects, such that they do not influence the executed signal

in the receding horizon framework.

> 7. Line 214: The use of $N_c$ as a "control horizon" referring to the number of actions that will be taken before the MPC problem is solved again is not the usual definition of "control horizon" in MPC literature (usually, "control horizon" refers to a subset of the prediction horizon where the control actions are allowed to change, while the MPC problem is still resolved at every "real" time step). It may be worth clarifying this.

Thank you for pointing this out, we now no longer refer to $N_c$ as a 'control horizon'.

> 8. Throughout, the authors highlight the capability of modeling secondary steering. However, I don't see any clear evidence that this modeling is needed; in fact, during steady-state periods, it seems that the MPC approach is essentially equivalent to the LUT in terms of power produced. Is the point here that waked turbines can achieve a satisfactory wake deflection with a smaller offset (due to the effective offset produced by the upstream wake), so loads may be kept lower? Perhaps this could be clarified.

The point of including secondary steering effects is indeed that the wake turbines can achieve the intended wake deflection with a smaller offset. Operating with a smaller yaw offset means that the waked turbines sacrifice less power for the same downstream gain. Loads are not mentioned in the current work, and might also play a role.

The LUT also implements secondary steering effects to reduce the commanded yaw offset. The near-equivalence in steady-state indicates that the secondary steering effects are properly implemented in the FVW controller.

> 9. Fig. 14 and paragraph starting at 355: The authors point out the differences in yaw offsets compared to the LUT approach due to the length of the horizon in the steady-state segments. Are these leading to a noticeable loss in power using the FVW controller?

Yes, there are the referenced steady-state segments do lead to a noticeable loss in power. This is visible in Figure 15, where the initial time segment shows underperformance of the FVW controller with respect to LUT controller. The paragraph starting at line 360 explains how the yaw offsets relate to this difference in power production.

> 10. Table 2: The different simulations seem to have quite different results in terms of yaw travel comparing LUT to FVW. Is this simply due to random variation? Why are results so different in different cases? Could you run more simulations or a longer simulation of the yawing behavior without LES, just to provide a clearer idea of the yaw activity? This is perhaps outside of the scope of the present work, but the yaw results are a little too varied to be convincing as it stands.

The difference in yaw travel between simulations is likely a result from the different wind direction signals, with each their unique transients and steady-state segments. Notably, the wind direction variation for TTWF and HKNA is synthetic for proof of concept and unlikely to occur in reality and the variation for HKNB and HKNC is a measured wind direction variation from separate days. The presented yaw travel results do support the conclusion that there is a possibility for yaw travel reduction. Future work might indicate whether this is robust under varying conditions and can be consistently achieved.

> 11. Line 392: The authors mention that the FVW is occasionally leading to oscillating

> yaw offsets. Shouldn't the R term in the MPC cost function limit this, or couldn't it be tuned to limit this behavior? As I understand it, $\psi(k_0 - 1)$ is the true previous offset applied.

This might indeed be addressed by tuning the weight $R$ in the objective function. However, that also limits responsiveness of the controller to variations in wind direction. We think a better solution would be development of a controller that combines dynamic, receding horizon control and information from steady-state, infinite horizon optimisation as we suggest in Section 5.4.

> I noted the following typos and very minor corrections:
> - Line 81: Please provide a reference for the Biot-Savart law

We have added a reference.

> - Line 105: Please provide the full form of $q_k$, as it is not clear to the reader exactly what makes up the model state

We have added a description of the composition of the full state vector.

> - I didn't see a wake deflection model—perhaps I missed this?

The wake of the free-vortex wake model naturally deflects under yaw-misaligned operation. There is no separate wake deflection model.

> - Line 137: "we a method" seems to be missing a word

Thanks for noticing, it has been adjusted to 'we present a method'.

> - Line 195: Are the splines turbine-specific? Or the same for all turbines? I do not quite understand the use of $n_b$. Is $n_b$ the number of degrees of freedom in a single spline? Are there multiple "splines" for one turbine? I am not very familiar with the terminology around splines.

The spline basis functions are the same for all turbines, where the number $n_\mathrm{b}$ is the number of basis functions. We have rephrased the introduction of this parameter to clarify. The coefficients that are multiplied with the basis functions to construct a yaw reference are turbine-specific.

> - Lines 226–227. The statement about the meaning of Q and R is repeated in this paragraph.

Thanks for noticing, we have removed the double sentence.

> - Line 250. Why is the more strict yaw controller needed to facilitate a fair comparison? Aren't all controllers using the same yaw logic regardless?

In an initial experiment, we found the results to be slightly inconsistent due to the length of time it took for the integrated error to reach the threshold specified by Kragh and Fleming (2012). This led to turbines operating way from their set-point for considerable time, especially for segments with constant wind direction. Lowering the threshold on the integrator yielded more consistent results

> - Line 293. I believe "LES" is used before the acronym is defined.

Line 25 introduced 'LES' as 'large-eddy simulation'.

- Fig. 12 caption: "turbine" → "turbines"

Adjusted.

- Line 421: "more optimal" is not very clearly defined, as formally, the value is either optimal or it is not. The authors could consider rephrasing this sentence.

We have rephrased the statement from 'can be more optimal' to 'can yield higher power production'.

- Lines 450–455: I appreciate the discussion of computational speed, however, I found these sentences difficult to follow. Please consider rephrasing.

We have adjusted the phrasing and hope that clarifies the discussion.

**References**

Grüne, L. and Pannek, J. (2017). *Nonlinear Model Predictive Control*. Springer, 2 edition.

Heck, K. S., Johlas, H. M., and Howland, M. F. (2023). Modelling the induction, thrust and power of a yaw-misaligned actuator disk. *J. Fluid Mech.*, 959:1–27.

Howland, M. F., González, C. M., Martínez, J. J. P., Quesada, J. B., Larrañaga, F. P., Yadav, N. K., Chawla, J. S., and Dabiri, J. O. (2020). Influence of atmospheric conditions on the power production of utility-scale wind turbines in yaw misalignment. *J. Renew. Sustain. Energy*, 12(6).

Kragh, K. A. and Fleming, P. A. (2012). Rotor Speed Dependent Yaw Control of Wind Turbines Based on Empirical Data. In *50th AIAA Aerosp. Sci. Meet.* AIAA.

Li, Z. and Yang, X. (2021). Large-eddy simulation on the similarity between wakes of wind turbines with different yaw angles. *J. Fluid Mech.*, 921:1–44.

Liew, J., Urbán, A. M., and Andersen, S. J. (2020). Analytical model for the power-yaw sensitivity of wind turbines operating in full wake. *Wind Energy Sci.*, 5(1):427–437.

van den Broek, M. J., De Tavernier, D., Hulsman, P., van der Hoek, D., Sanderse, B., and van Wingerden, J. W. (2023). Free-vortex models for wind turbine wakes under yaw misalignment – a validation study on far-wake effects. *Wind Energy Sci. Discuss.*, 2023:1–27.